# Transport of DNA within cohesin involves clamping on top of engaged heads by Scc2 and entrapment within the ring by Scc3

James E Collier[1†], Byung-Gil Lee[2†], Maurici Brunet Roig[1], Stanislav Yatskevich[1], Naomi J Petela[1], Jean Metson[1], Menelaos Voulgaris[1], Andres Gonzalez Llamazares[2], Jan Löwe[2]*, Kim A Nasmyth[1]*

[1]Department of Biochemistry, University of Oxford, Oxford, United Kingdom; [2]MRC Laboratory of Molecular Biology, Cambridge, United Kingdom

*For correspondence:
jyl@mrc-lmb.cam.ac.uk (JL);
ashley.nasmyth@bioch.ox.ac.uk (KAN)

†These authors contributed equally to this work

Competing interests: The authors declare that no competing interests exist.

**Abstract** In addition to extruding DNA loops, cohesin entraps within its SMC-kleisin ring (S-K) individual DNAs during G1 and sister DNAs during S-phase. All three activities require related hook-shaped proteins called Scc2 and Scc3. Using thiol-specific crosslinking we provide rigorous proof of entrapment activity in vitro. Scc2 alone promotes entrapment of DNAs in the E-S and E-K compartments, between ATP-bound engaged heads and the SMC hinge and associated kleisin, respectively. This does not require ATP hydrolysis nor is it accompanied by entrapment within S-K rings, which is a slower process requiring Scc3. Cryo-EM reveals that DNAs transported into E-S/E-K compartments are 'clamped' in a sub-compartment created by Scc2's association with engaged heads whose coiled coils are folded around their elbow. We suggest that clamping may be a recurrent feature of cohesin complexes active in loop extrusion and that this conformation precedes the S-K entrapment required for sister chromatid cohesion.

## Introduction

Protein complexes containing SMC and kleisin subunits organise the spatial arrangement, or topology, of DNAs in most if not all living organisms (*Nasmyth, 2001*; *Yatskevich et al., 2019*). Best characterised are the eukaryotic cohesin and condensin complexes that are thought to organise chromosomal DNAs during interphase and mitosis, respectively, by a process of loop extrusion (LE) (*Golfier et al., 2020*). Cohesin in addition mediates the connections between sister DNAs that hold sister chromatids together during mitosis until their disjunction at the onset of anaphase (*Oliveira et al., 2010*; *Uhlmann et al., 1999*). Many clues as to their molecular mechanisms have emerged from structural studies. All contain a pair of rod-shaped SMC proteins with a dimerisation domain, known as the hinge, at one end and an ABC-like ATPase domain at the other, separated by a ~ 50 nm long anti-parallel intra-molecular coiled coil (*Haering et al., 2002*). Their association creates V-shaped dimers whose apical ATPase head domains are interconnected by a kleisin subunit (Scc1) whose N-terminal domain forms a three-helix bundle with the coiled coil emerging from Smc3's ATPase head (*Gligoris et al., 2014*), called its neck, and whose C-terminal winged helical domain binds to the base (or cap) of Smc1's ATPase head to complete the ring (*Haering et al., 2004*).

Hinge dimerisation facilitates numerous other contacts between Smc1 and Smc3. First, their coiled coils interact with each other extensively, all the way from the hinge to the joint, a small break in the coiled coil roughly 5 nm above the heads, effectively zipping up the coiled coils (*Bürmann et al., 2019*; *Chapard et al., 2019*; *Diebold-Durand et al., 2017*; *Soh et al., 2015*). This

process leads to the juxtaposition of the Smc1 and Smc3 heads, which are loosely associated under these conditions (*Chapard et al., 2019*; *Diebold-Durand et al., 2017*). Second, their coiled coils fold around an elbow, which results in an interaction between the hinge and a section of the coiled coils approximately 10 nm from the heads (*Bürmann et al., 2019*). Finally, the γ-phosphate of ATP bound to one ATPase head binds a signature motif on the other, resulting under appropriate conditions in engagement of the heads and a sandwiching of two molecules of ATP between them, a process that is a precondition for subsequent ATP hydrolysis (*Arumugam et al., 2003*; *Lammens et al., 2004*; *Íñigo et al., 2017*). Head engagement has been proposed to disrupt coiled coil interactions, at least in the vicinity of the heads, yet the full extent of this disruption is not known.

The ATPase activities of SMC-kleisin complexes as well as all their biological functions in vivo depend on additional proteins that are recruited through their association with kleisin subunits and act by binding DNA and interacting with various SMC protein domains. In cohesin, this class of proteins consists of large hook-shaped proteins composed of HEAT repeats, known as Heat repeat containing proteins Associated With Kleisins (HAWKs) (*Wells et al., 2017*). Cohesin has three such HAWKs known as Scc2, Scc3, and Pds5. Scc3 is thought to be permanently bound to the complex (*Tóth et al., 1999*) while association of Scc2 and Pds5, whose occupancy is mutually exclusive, are more dynamic (*Petela et al., 2018*). Scc2 is essential for cohesin's ATPase activity (*Petela et al., 2018*), for its loading onto chromosomes (*Ciosk et al., 2000*), for maintaining cohesin's chromosomal association during G1 (*Srinivasan et al., 2019*), and for cohesin's ability to extrude loops in vitro (*Davidson et al., 2019*; *Kim et al., 2019*). However, Scc2 is not required to maintain cohesion during G2 or even establish cohesion during S phase from complexes previously associated with unreplicated DNAs (*Srinivasan et al., 2019*). Pds5 also has multiple functions. By recruiting Wapl, it promotes cohesin's dissociation from chromosomes, a process blocked by acetylation of two lysine residues on Smc3 during S phase (*Beckouët et al., 2016*; *Chan et al., 2013*; *Chan et al., 2012*). Pds5 also promotes acetylation during S phase and inhibits deacetylation during G2 and thereby protects sister chromatid cohesion which would otherwise be destroyed by Wapl-mediated release (*Chan et al., 2013*).

Though the mechanism by which cohesin extrudes loops remains mysterious, there is a clear and simple hypothesis as to how cohesin holds sister DNAs together, namely by entrapping them both inside the S-K ring created through the binding of a kleisin subunit to the ATPase heads of an Smc1/Smc3 heterodimer (*Gruber et al., 2003*; *Haering et al., 2002*). This model explains the key observation that cleavage of cohesin's kleisin subunit by separase, or any other site-specific protease, is sufficient to trigger sister chromatid disjunction at anaphase (*Oliveira et al., 2010*; *Uhlmann et al., 2000*). To measure such entrapment in yeast, we have substituted residues within all three interfaces that make up S-K rings by pairs of cysteine residues that can be crosslinked by the thiol-specific reagent bis-maleimidoethane (BMOE). Around 20% of cohesin complexes can be crosslinked simultaneously at all three interfaces in vivo (*Gligoris et al., 2014*), and in post-replicative cells this is accompanied by formation of SDS-resistant structures that hold together the sister DNAs of circular minichromosomes, called catenated dimers or CDs (*Chapard et al., 2019*; *Gligoris et al., 2014*; *Srinivasan et al., 2018*). Because the two DNAs associated with CDs are not otherwise intertwined (*Haering et al., 2008*), they must be held together by cohesin through a topological mechanism, either by co-entrapment within a chemically circularised S-K ring or conceivably in a three-way Borromean ring containing a pair of sister DNA rings and an SDS-resistant S-K ring. Importantly, the study of numerous mutants has revealed a perfect correlation between CD formation and cohesion establishment (*Srinivasan et al., 2018*), suggesting that these structures are actually responsible for sister chromatid cohesion or at the very least are produced by a highly related mechanism. Using cysteine pairs that crosslink heads that are not engaged, but are otherwise closely juxtaposed (J) (*Chapard et al., 2019*), it has been established that sister DNAs are at least some of the time entrapped between juxtaposed heads and the kleisin associated with them, namely within a J-K subcompartment of the S-K ring (see *Figure 1A* for an overview of the compartments).

Loading of cohesin onto minichromosomes during G1 leads to a different topological interaction, namely catenation of individual circular DNAs by S-K rings (chemically circularised for detection), known as catenated monomers or CMs. Though loading of cohesin throughout the genome is normally accompanied by CM formation on minichromosomes, cohesin complexes containing a hinge with mutations within its lumen that neutralises its positive charge can load throughout the genome but cannot form either CMs or CDs (*Srinivasan et al., 2018*), implying that stable chromosomal

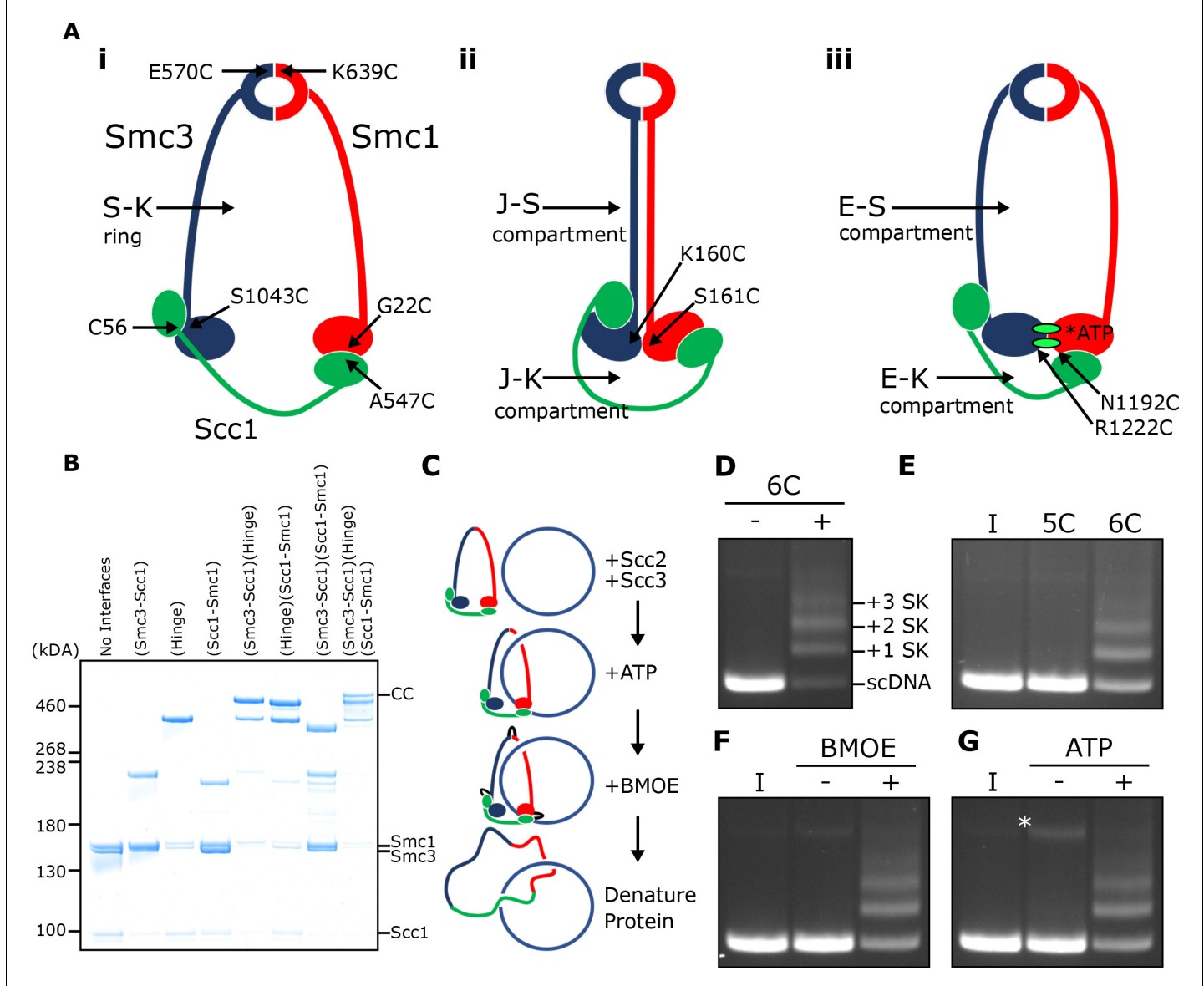

**Figure 1.** SMC-kleisin (S–K) rings entrap circular DNA in vitro. (A) Cohesin's different compartments and the position of cysteine pairs used in our crosslinking studies. (B) BMOE-induced crosslinking of S-K rings with cysteine pairs in the specified interfaces. CC = circular cohesin. (C) The entrapment assay scheme. (D) Entrapment of DNA in S-K rings in the presence or absence of 6C cohesin, or (E) the presence 5C cohesin, lacking Scc1A547C, or 6C cohesin. (F) For DNA in the presence of Scc2, Scc3, and 6C cohesin and the presence or absence of BMOE, or (G) the presence or absence of ATP. Entrapment assays incubated for 40 min (*=damaged open circular DNA; I = input DNA).

The online version of this article includes the following figure supplement(s) for figure 1:

**Figure supplement 1.** Related to *Figure 1*.

association is not necessarily synonymous with entrapment of DNAs within S-K rings. That cohesin can associate with DNA in a functional manner without being topologically entrapped within S-K rings is supported by the finding that LE in vitro can be mediated by a version of human cohesin whose Smc1, Smc3, and kleisin subunits are expressed as a single polypeptide and whose hinge interface has been crosslinked by BMOE (*Davidson et al., 2019*). Importantly, neither of the above observations exclude the possibility that cohesin usually associates with chromosomal DNAs by entrapping a loop within its S-K ring, a type of association that has been termed pseudo-topological.

Previous reports claiming entrapment of DNA within cohesin rings in vitro used salt resistance and sensitivity to cleavage as their criteria (*Murayama and Uhlmann, 2015*; *Murayama and Uhlmann, 2014*). However, there are fundamental limitations to such experiments. Many types of association other than entrapment within a closed compartment could give rise to salt resistance and cleavage sensitivity. Equally serious, even if these criteria were indicative of topological entrapment, they reveal little or nothing as to its nature, namely whether DNAs are entrapped in S-K rings or other closed compartments, for example the E-S and E-K compartments between ATP-bound engaged heads and the SMC hinge and associated kleisin, respectively, or indeed other types of compartment created by multiple contacts between HAWKs and SMC proteins. For these reasons, we describe here the use of thiol-specific crosslinking to measure bona fide topological entrapment of DNAs within S-K rings in vitro. Both Scc2 and Scc3 are essential for this process, as are their abilities to bind DNA. The process is dependent on ATP binding and stimulated by its hydrolysis, a feature largely absent from previous assays (*Minamino et al., 2018*).

Remarkably, we find that Scc2 alone promotes the rapid entrapment of DNAs within E-S and E-K compartments in a process that is not accompanied by entrapment within S-K rings, and propose that E-S/E-K entrapment occurs simultaneously through a single mechanism. Because E-S/E-K entrapment is an order of magnitude more rapid than S-K entrapment, we suggest that creation of the former by Scc2 may be a precursor to the latter, a process contingent on the action of Scc3. Electron cryo-microscopy (cryo-EM) of complexes formed between cohesin's SMC-kleisin trimers and linear or circular DNAs in the presence of Scc2 suggests that entrapment within E-S/E-K compartments involves transport of DNA between ATPase heads prior to their engagement, whereupon DNAs are 'clamped' in a sub compartment formed by Scc2's association with engaged heads in a manner similar to that recently observed in a complex between DNA and both human and *Schizosaccharomyces pombe* cohesin associated with both Scc2[NIPBL/Mis4] and Scc3[SA2/Psc3] (*Higashi et al., 2020*; *Shi et al., 2020*). Our observations reveal key insights into the biochemical activities of Scc2 and Scc3 and suggest that the recurrent clamping of DNAs by Scc2[NIPBL/Mis4] and engaged heads resulting in E-S/E-K entrapment, followed by their subsequent release, may be an integral aspect of cohesin's ability to load onto and translocate along DNA.

## Results

### SMC-kleisin rings entrap circular DNA in vitro

We expressed cohesin trimers from *Saccharomyces cerevisiae* consisting of Smc1, Smc3, and Scc1 in insect cells using the baculovirus expression system (*Figure 1—figure supplement 1A*). Scc3 was expressed separately because co-expression with trimers resulted in substoichiometric yields. We also expressed a version of Scc2 lacking its N-terminal domain (Scc2C, residues 133–1493). Though this form no longer binds Scc4, it is fully capable of activating cohesin's ATPase activity (*Petela et al., 2018*), and for simplicity we will refer to this as Scc2 throughout most of the text. To measure entrapment of DNAs inside S-K rings (*Figure 1A* i), we introduced cysteine pairs within all three ring interfaces (Smc1K639C-Smc3E570C, Smc1G22C-Scc1A547C, and Smc3S1043C-Scc1C56) that enables them to be crosslinked using BMOE. Individual interfaces were crosslinked with efficiencies varying from 30–70% and by comparing the migration of proteins following crosslinking cysteine pairs at single (2C), double (4C), and triple (6C) interfaces, we identified a covalently circular species only produced when all three interfaces were crosslinked (*Figure 1B*).

To measure DNA entrapment within cohesin's S-K compartment in vitro, 6C SMC-kleisin trimers were mixed with circular supercoiled DNAs, Scc2, and Scc3, and incubated for 40 min at 24°C following addition of ATP. BMOE was then added and the mixture placed on ice for 6 min after which proteins were denatured by adding SDS to a final concentration of 1% and heating at 70°C for 20 min. The DNA was then fractionated by agarose gel electrophoresis and visualised by ethidium bromide staining (*Figure 1C & D*). Addition of 6C trimers to Scc2/Scc3/DNA mixtures greatly reduced the amount of DNA co-migrating with supercoiled monomers and produced a ladder of retarded DNA species, most likely caused by successive entrapment by one, two, three and more S-K rings (*Figure 1D*).

We propose that the ladder corresponds to multiple cohesin rings entrapping individual DNAs and not entrapment of multiple DNAs by individual cohesin rings for two reasons. First, retardation

caused by entrapment within E-S compartments (see below), which contain only Smc1 and Smc3, is less than that caused by entrapment within S-K or E-K compartments, which contain Scc1 as well as Smc1 and Smc3 (see Figure 5B). Second, dimeric plasmid DNA, which is frequently present in plasmid preparations, although largely absent from these gels due to our purification protocol, runs roughly at the top of the gel with respect to our figures. Thus, if our ladders represented entrapment of multiple DNAs by individual cohesin rings, the DNA retardation should be much greater. Those DNAs retarded by entrapment within a single ring correspond to the CMs previously observed in vivo (*Gligoris et al., 2014*). Ladder formation required cysteine pairs at all three interfaces. It was never observed with linear DNA (*Figure 1—figure supplement 1D*) or when just a single cysteine (*Figure 1E*) or BMOE (*Figure 1F*) was omitted. Crucially, the ladders were strictly dependent on addition of ATP (*Figure 1G*).

## Entrapment of DNAs by S-K rings requires Scc3 and is stimulated by Scc2

To assess the roles of Scc2 and Scc3, we measured ladder formation at four successive 10 min intervals in the presence and absence of the two proteins. Ladders indicative of entrapment increased with time (up to 40 min), suggesting that formation is a slow process, were greatly reduced by omission of Scc2 (*Figure 2A*), and almost completely abolished by omission of Scc3 (*Figure 2B*). In the absence of both Scc2 and Scc3, the level of entrapment was comparable to that observed in the presence of Scc2 alone (data not shown).

## Entrapment of DNAs by S-K rings depends on ATP binding to Smc3 and on ATP hydrolysis

To address the role of cohesin's ATPase, we mutated Smc3's Walker A site (Smc3K38I) to abolish ATP binding to Smc3. This almost completely abolished entrapment (*Figure 2C*). We did not test the effect of mutating the equivalent residue in Smc1 as this has previously been shown to abolish association of Smc1/3 heterodimers with Scc1 (*Arumugam et al., 2003*). We next tested the effect of mutating both Walker B sites to residues that permit ATP binding but strongly inhibit hydrolysis (Smc1E1158Q Smc3E1155Q, 'EQEQ') (*Figure 1—figure supplement 1B*), which caused a more modest, albeit still significant, reduction (*Figure 2D*). These data suggest that cohesin's ability to complete the ATP hydrolysis cycle stimulates entrapment but is not strictly necessary. To address whether Smc3's K112 K113 are also important we analysed the effect of substituting them by glutamine (Smc3K112Q K113Q), mutations thought to mimic the acetylated state. This also reduced S-K entrapment (*Figure 1—figure supplement 1C*), an effect that parallels its abrogation of cohesin loading in vivo (*Hu et al., 2015*).

## DNA binding to Scc3 is required for its entrapment by S-K rings

During a search for cohesin domains that bind DNA, we discovered that Scc3's association with a fragment of Scc1 containing residues 269–451 greatly stimulates its association with double stranded DNA, as measured using an electrophoretic mobility shift assay (EMSA) (*Figure 3—figure supplement 1A*). Reasoning that Scc3/Scc1 complexes might bind DNA in a similar manner to that recently observed in a co-crystal of DNA bound to condensin's Ycg1 HAWK bound to its kleisin partner Brn1 (*Kschonsak et al., 2017*), we mutated two clusters of positively charged residues (Scc3K224E K225E R226E and Scc3 K423E K513E K520E) on opposite sides of the groove within Scc3 that is equivalent to Ycg1's DNA binding groove (*Figure 3A*). Neither triple (3E) mutant eliminated DNA binding (*Figure 3—figure supplement 1B*) nor caused lethality (*Figure 3—figure supplement 1E*). Despite this, both reduced cohesin's association with all genomic sequences except point centromeres (*CEN*s) (*Figure 3—figure supplement 1C & D*). In contrast, combining the two triple mutations (to create 6E) was lethal (*Figure 3—figure supplement 1E*), abolished binding of Scc3 to DNA in the presence of Scc1 (*Figure 3B*), and with the exception of *CEN*s eliminated cohesin's association with the genome (*Figure 3C & D*).

Remarkably, cohesin containing Scc3-6E accumulated to exceptionally high levels at *CEN*s (*Figure 3C & D*), which are the loading sites for most peri-centric cohesin (50 kb intervals surrounding *CEN*s). This distribution resembles that of Scc2 in wild type cells and indeed, *scc3-6E* had little or no effect on Scc2's accumulation with *CEN*s (*Figure 3E*). This implies that cohesin containing

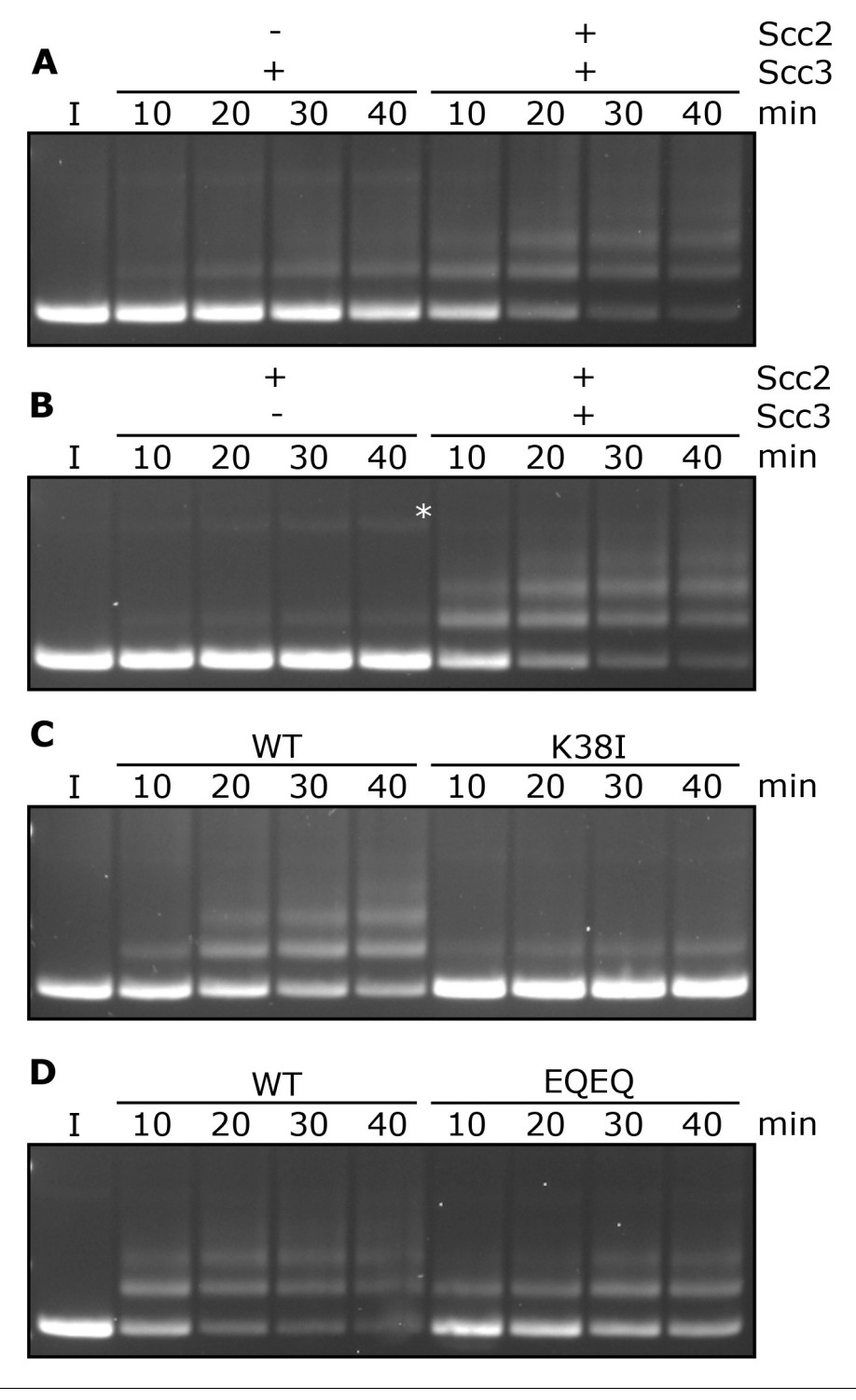

**Figure 2.** Entrapment within S-K rings requires both Scc2 and Scc3, ATP binding to Smc3, and is stimulated by ATP hydrolysis. (**A**) Entrapment of DNA in S-K rings in the presence of Scc3, and the presence or absence of Scc2, or (**B**) the presence of Scc2, and the presence or absence of Scc3 (*=damaged open circular DNA). (**C**) DNA entrapment in the presence of Scc2 and Scc3, comparing WT cohesin to Smc3K38I (K38I), or (**D**) WT cohesin to Smc1E1158Q Smc3E1155Q (EQEQ). Entrapment assays incubated for 40 min with time points taken every 10 min (I = input DNA).

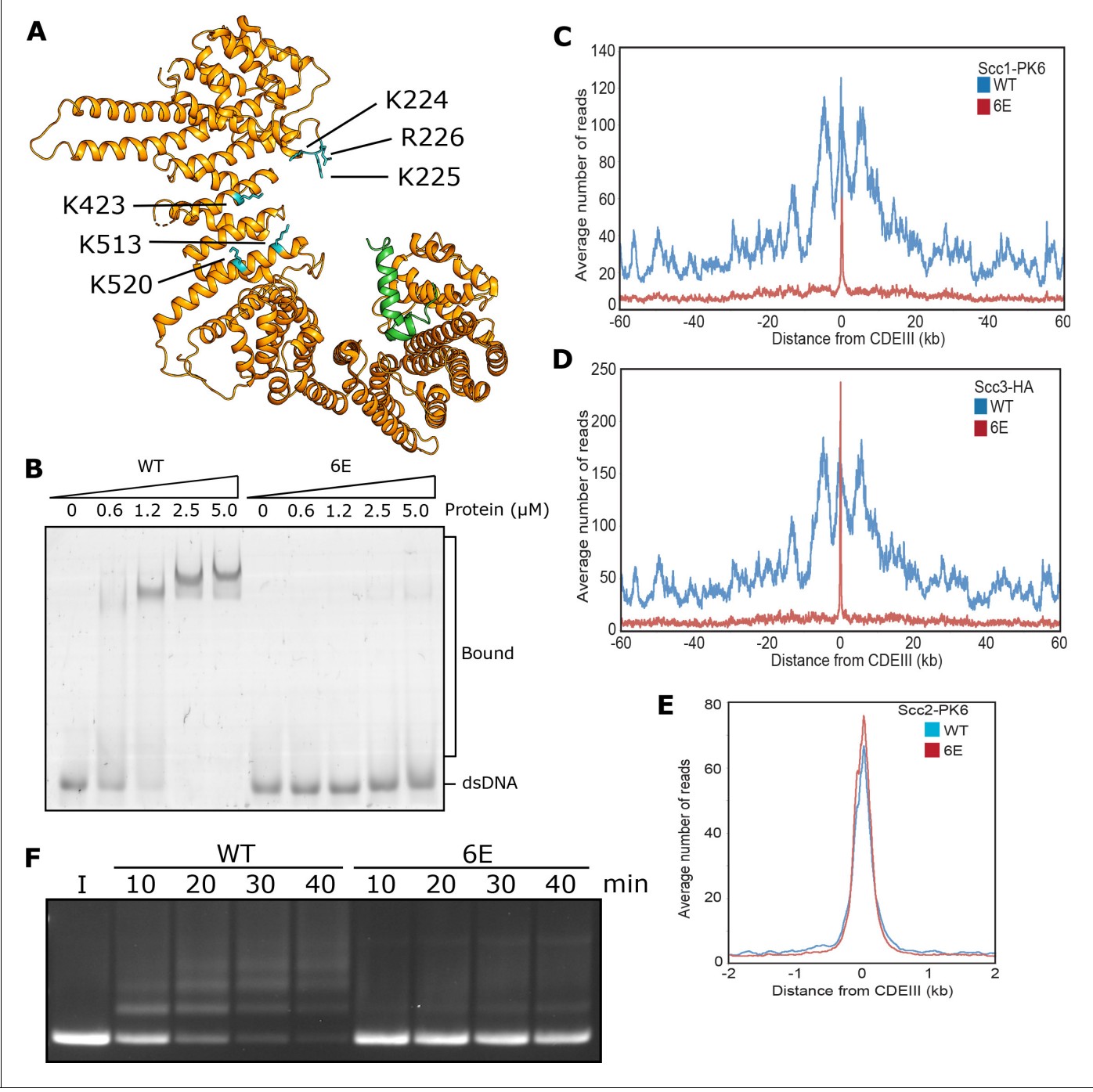

**Figure 3.** DNA binding to Scc3 is required for its entrapment by S-K rings. (**A**) Structure of *S. cerevisiae* Scc3 (orange) protein in complex with a fragment of Scc1 (green) (PDB 6H8Q). Labelled are the six residues within the DNA binding groove of Scc3 that were mutated to glutamate (Scc3-6E). (**B**) EMSA comparing the ability of WT Scc3-Scc1$^{269-451}$ and Scc3-6E-Scc1$^{269-451}$ complexes to bind dsDNA. (**C**) Average calibrated ChIP-seq profiles of Scc1-PK6 60 kb either side of *CEN*s in the presence of ectopic WT Scc3 (KN27821) or Scc3-6E (KN27804). Cells were arrested in G1 with α-factor prior to release into auxin and nocodazole containing media at 25°C to deplete the endogenous Scc3. ChIP-seq samples were taken 60 min after release. (**D**) Average calibrated ChIP-seq profile of ectopic WT (KN27796) or mutant (KN27802) Scc3-HA performed as in C. (**E**) Average calibrated ChIP-seq profile of Scc2-PK6 in the presence of ectopic WT Scc3 (KN28075) or Scc3-6E (KN28287). Experiment was performed as in C. (**F**) Entrapment of DNA within S-K rings in the presence Scc2 and either WT Scc3 or Scc3-6E. Entrapment assay incubated for 40 min with time points taken every 10 min (I = input DNA). The online version of this article includes the following figure supplement(s) for figure 3:

**Figure supplement 1.** Related to *Figure 3*.

Scc3-6E forms complexes with Scc2 at *CEN*s but subsequently fails to form a stable association with chromatin or translocate into neighbouring sequences. Our ability to detect such complexes at *CEN*s but not at other loading sites along chromosome arms can be attributed to the fact that Scc2's partner Scc4 binds to the kinetochore protein Ctf19 and this association transiently tethers complexes at *CEN*s while they are attempting to load (*Hinshaw et al., 2017*). Though accumulation of cohesin bound by Scc2 at *CEN*s does not depend on Scc3's ability to bind DNA, it does still require Scc3 (*Figure 3—figure supplement 1F*). Crucially, cohesin containing Scc3-6E failed to support entrapment of DNAs inside S-K rings in vitro (*Figure 3F*). During the course of our work, a crystal structure of DNA bound to a Scc3/Scc1 complex confirmed that it does indeed bind DNA (*Li et al., 2018* PDB 6H8Q) in a manner resembling that of Ycg1. Moreover, K224, K225 R226, K423, K513, and K520 are all predicted to contribute to the association. These data imply that Scc3's ability to bind DNA has an important role in cohesin's ability to load onto and translocate along chromosomal DNA in vivo, as well as entrap in S-K rings in vitro.

## DNA binding to Scc2 facilitates entrapment by S-K rings

The *S. pombe* Scc2/4 complex has previously been shown to bind DNA in vitro (*Murayama and Uhlmann, 2014*) but the physiological significance of this activity has never been investigated. EMSA revealed that *S. cerevisiae* Scc2 also binds DNA (*Figure 4A*), as do Scc2/4 complexes with slightly higher affinity (*Figure 4—figure supplement 1A*). Unlike Scc3, whose DNA binding was greatly enhanced by Scc1, DNA binding by Scc2 was reduced by addition of a Scc1 fragment (Scc1$^{150-298}$) that contains sequences necessary for Scc2-dependent loading in vivo (*Figure 4A*; *Petela et al., 2018*). Interestingly, the inhibitory effect of Scc1$^{150-298}$ was not observed in the binding of DNA to full length Scc2/4 (*Figure 4—figure supplement 1A*), suggesting that DNA binding sites also exist in Scc4, or in sequences N-terminal of the deletion in our Scc2C construct.

An alignment of the crystal structure of *E. gossypii* Scc2 (*Chao et al., 2017*) with that of Ycg1/ Brn1bound to DNA (*Kschonsak et al., 2017*) revealed not only a remarkable similarity in the overall shape of their hook-shaped HEAT repeats but also a set of potential DNA binding residues on the surface of the shallow concave groove corresponding to Ycg1's DNA binding pocket (*Petela et al., 2018*; *Figure 4B*). Four of these are particularly conserved and correspond to S717, K721, K788, and H789 in *S. cerevisiae*. Both *scc2S717L K721E* and *scc2K788E H789E* double mutants are lethal and abolish loading of cohesin throughout most of the genome (*Petela et al., 2018*; *Figure 4—figure supplement 1C*). To test whether these residues participate in binding DNA, we used EMSAs to measure the effect on DNA binding of mutating the above residues to glutamate. Both Scc2S717E K721E and Scc2K788E H789E double mutants reduced binding (*Figure 4—figure supplement 1B*) but caused only a modest reduction in S-K entrapment (data not shown). In contrast, the quadruple mutant Scc2S717E K721E K788E H789E (Scc2-4E) not only greatly reduced DNA binding (*Figure 4C*) but also S-K entrapment (*Figure 4D*). These results suggest that Scc2's ability to bind DNA has a crucial role in entrapping DNA within S-K rings in vitro, an activity also required for loading cohesin onto chromosomes in vivo (*Petela et al., 2018*; *Figure 4—figure supplement 1C*). They also demonstrate that the stimulation of DNA entrapment within S-K rings by Scc2 is not merely an adventitious property of Scc2 but an activity dependent on conserved surface residues that have unambiguous physiological functions.

## DNA is never entrapped in J-S and only rarely in J-K compartments

Using the Smc1S161C-Smc3K160C cysteine pair, cohesin's ATPase heads can also be efficiently crosslinked in the J-state (*Chapard et al., 2019*). Moreover, this crosslinking can be combined with simultaneous crosslinking of N- and C-terminal kleisin domains to Smc3 and Smc1 ATPase heads respectively, to measure entrapment within J-K compartments (*Figure 5—figure supplement 1A*), or with simultaneous crosslinking of the hinge (*Figure 5—figure supplement 1B*), to measure entrapment of DNAs in J-S compartments (*Figure 1A* ii). J crosslinking alone or in combination with hinge (J-S) or kleisin (J-K) was efficient even in the presence of ATP, DNA, Scc2, and Scc3 (*Figure 5— figure supplement 1C*). In other words, both J-S and J-K circularisation occurred efficiently under conditions that promote efficient entrapment of DNAs inside S-K rings. However, DNAs were never entrapped within J-S compartments and only rarely by J-K ones (*Figure 5—figure supplement 1D*). J-K entrapment was not only much less frequent than S-K entrapment but also independent of Scc2.

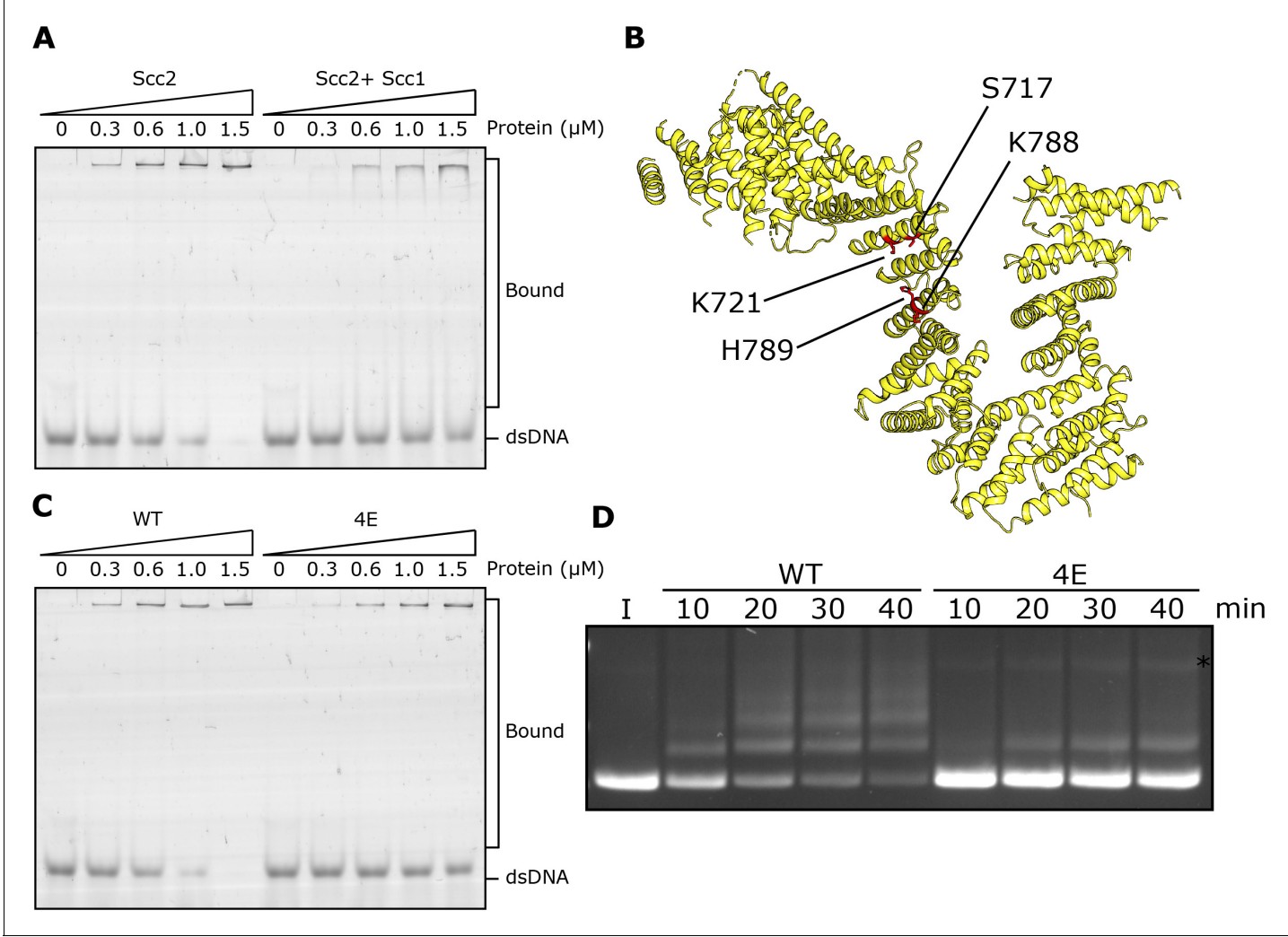

**Figure 4.** DNA binding to Scc2 facilitates entrapment by S-K rings. (**A**) EMSA comparing the ability of Scc2 and Scc2-Scc1[150-298] complexes to bind dsDNA. (**B**) *S. cerevisiae* Scc2 from the cryo-EM structure (*Figure 8*) with the four resides within the putative DNA binding surface labelled that were mutated to glutamate (Scc2-4E). (**C**) EMSA comparing the ability of Scc2 and Scc2-4E complexes to bind dsDNA. (**D**) Entrapment of DNA in S-K rings in the presence of Scc3 and either Scc2 or Scc2-4E. Entrapment assay incubated for 40 min with time points taken every 10 min (*=damaged open circular DNA; I = input DNA).

The online version of this article includes the following figure supplement(s) for figure 4:

**Figure supplement 1.** Related to *Figure 4*.

The fact that J-K entrapment was comparable to S-K entrapment in the absence of Scc2 (compare *Figures 2A* and *Figure 5—figure supplement 1D*) suggests that the low-level entrapment of DNAs in S-K rings induced by Scc3 alone may in fact correspond to DNAs entrapped in J-K compartments. Though J-K circularisation by BMOE is modestly lower than that of S-K, this cannot account for its far lower DNA entrapment. We therefore suggest that most ATPase heads associated with DNA entrapped within S-K rings in vitro are not juxtaposed. They are either fully disengaged, in the E-state, or in some other conformation.

## Rapid DNA entrapment in E-S and E-K compartments

We used the same approach to measure entrapment in E-S or E-K compartments (*Figure 1A* iii), in this case replacing J-specific cysteines by a pair specific for the E-state (Smc1N1192C-Smc3R1222C). Unlike J crosslinking, which was readily detected in cohesin trimers, efficient E-state crosslinking was dependent on the presence of ATP (*Figure 5—figure supplement 2A*). As with J-, E-state

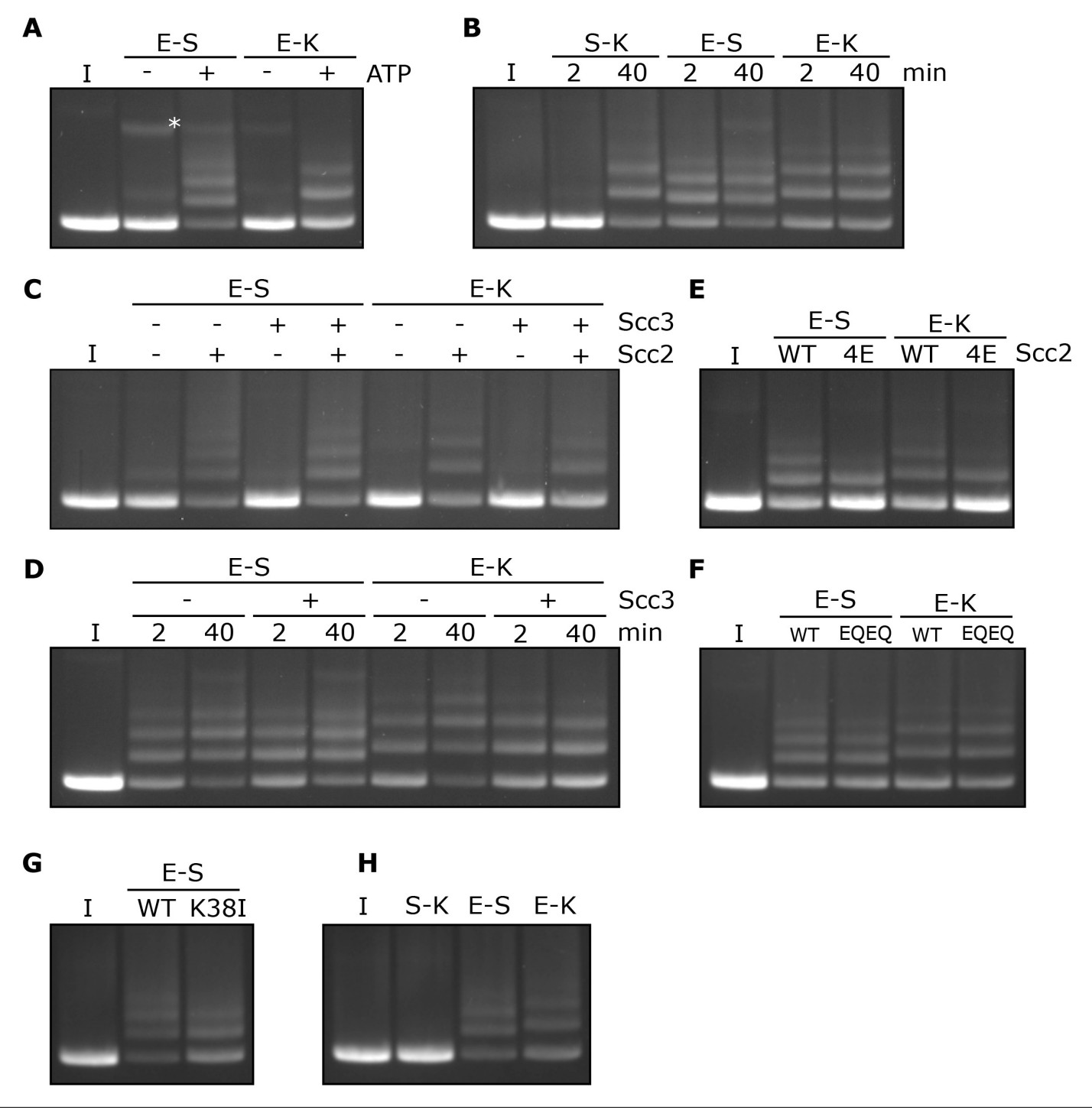

**Figure 5.** Rapid DNA entrapment in E-S and E-K compartments. (A) Entrapment of DNA in E-S/E-K compartments in the presence of Scc2 and Scc3, and the presence or absence of ATP, incubated for 40 min (*=damaged open circular DNA). (B) DNA entrapment in S-K rings, or E-S/E-K compartments in the presence of Scc2 and Scc3, incubated for either 2 or 40 min. (C) DNA entrapment in E-S/E-K compartments in the presence of Scc2, Scc3, Scc2 and Scc3, or absence of both, incubated for 2 min. (D) DNA entrapment in E-S/E-K compartments in the presence of Scc2, and either the presence or absence of Scc3, incubated for either 2 min or 40 min. (E) DNA entrapment in E-S/E-K compartments in the presence of either Scc2 or Scc2-4E, or (F) Entrapment in the presence of Scc2 alone, comparing WT and Smc1E1158Q Smc3E1155Q (EQEQ) cohesin, incubated for 2 min. (G) DNA entrapment in E-S compartments in the presence of Scc2 comparing WT and Smc3K38I (K38I) cohesin. (H) Entrapment of DNAs in S-K rings, or E-S/E-K compartments, in the presence of Scc2, incubated for 2 min (I = input DNA).

The online version of this article includes the following figure supplement(s) for figure 5:

*Figure 5 continued on next page*

*Figure 5 continued*

**Figure supplement 1.** DNA is never entrapped in J-S and only rarely in J-K compartments.

**Figure supplement 2.** Circularisation of the E-S and E-K compartments and E- and J-state crosslinking under different conditions.

crosslinking can be combined with simultaneous crosslinking of N- and C-terminal kleisin domains to Smc3 and Smc1 ATPase heads respectively, to measure entrapment within E-K compartments (*Figure 5—figure supplement 2B*), or with simultaneous crosslinking of the hinge (*Figure 5—figure supplement 2C*), to measure entrapment of DNAs in E-S compartments. As previously reported (*Chapard et al., 2019*), Smc1/3 dimers crosslinked simultaneously at the hinge and engaged heads co-migrate with those crosslinked at the hinge alone, which hinders detection of E-S circularisation directly. Given that double crosslinking has been detected in vivo using differently tagged proteins (*Chapard et al., 2019*) and that DNAs are readily entrapped by Smc1/3 dimers containing hinge and E-specific cysteine pairs treated with BMOE, we can reason that efficient double crosslinking does indeed occur. DNAs were entrapped in an ATP-dependent fashion in both E-S and E-K compartments in the presence of Scc2 and Scc3 (*Figure 5A*). Notably, both processes occurred much more rapidly than S-K entrapment, with significant amounts of DNA entrapped by multiple rings within 2 min (*Figure 5B*). Because S-K entrapment occurs much more slowly, the efficient entrapment of DNAs inside E-S/E-K compartments within a few minutes is presumably not accompanied by S-K entrapment. Though it occurs efficiently in vitro, entrapment of circular DNAs by cohesin in E-S compartments has not so far been detected in vivo, although *Bacillus subtilis* SMC possessing Walker B mutations have been shown to have such an activity inside cells (*Vazquez Nunez et al., 2019*).

## Entrapment within the E-S and E-K compartments depends on Scc2 but not Scc3

In contrast to entrapment within S-K rings, which depends on Scc3, the rapid entrapment of DNA in the E-S/E-K compartments was Scc3 independent (*Figure 5C*). However, it was highly dependent on Scc2, both in the presence or absence of Scc3 (*Figure 5C*). Levels of E-S entrapment increased between 2 and 40 min in the presence of Scc2, as well as in the presence of Scc2 and Scc3 (*Figure 5D*). However, while a similar result was seen for E-K entrapment in the presence of Scc2 alone, this increase was not observed in the presence of both Scc2 and Scc3 which instead showed no increase by the longer time point and possibly even a small reduction.

The rapid entrapment of DNAs within E-S/E-K compartments in the presence of Scc2 was reduced by Scc2-4E (*Figure 5E*), suggesting that the reaction at least partly depends on Scc2's ability to bind DNA. Strikingly, both types of entrapment were unaffected by Smc1E1158Q Smc3E1155Q mutations (EQEQ), implying that neither form of entrapment requires ATP hydrolysis (*Figure 5F*). In contrast to S-K entrapment in the presence of both Scc2 and Scc3 (*Figure 2C*), E-S entrapment in the presence of Scc2 alone was only modestly reduced by Smc3K38I, implying that ATP bound merely to Smc1's ATPase head is sufficient (*Figure 5G*). Indeed, Smc3K38I does not prevent E-specific crosslinking under these reaction conditions, namely in the presence of ATP, DNA, and Scc2 (*Figure 5—figure supplement 2D*) and its modest reduction of E-S entrapment is in line with its effect on E-state crosslinking (*Figure 5G* and *Figure 5—figure supplement 2D*). Though it does not abolish head engagement, Smc3K38I clearly compromises the process. As long as ATP is present, neither DNA nor Scc2 are required for efficient E-state crosslinking of wild type complexes but both are important for Smc3K38I complexes (*Figure 5—figure supplement 2D*). Smc3K38I presumably destabilises head engagement in a manner that can be overcome by the presence of Scc2 and DNA. To explore whether Scc2 and DNA also promote head engagement of otherwise wild type complexes, we tested their effect when ATP's ability to promote head engagement is compromised by omission of $Mg^{2+}$ (*Figure 5—figure supplement 2E*). Under these circumstances, addition of both Scc2 and DNA restored efficient head engagement and both factors were required for this effect. Our finding that Scc2 and DNA collaborate to promote ATP-dependent head engagement suggests that DNA binds to a site created by head engagement as well as to Scc2.

## Scc2 causes DNAs to be entrapped in E-S and E-K compartments without entering S-K rings

Because Scc3 is crucial for S-K entrapment, the rapid entrapment of DNAs within E-S and E-K compartments in the presence of Scc2 alone should be unaccompanied by S-K entrapment. This is indeed the case. In contrast with E-S or E-K entrapment, which is very efficient, few if any DNAs are entrapped in S-K compartments by 2 min (*Figure 5H*). Though paradoxical, this striking observation has a very simple explanation. The similarity in kinetics suggests that E-S and E-K entrapments are created simultaneously as part of the same reaction. In other words, a single type of DNA passage followed by head engagement gives rise to both types. We envisage two types of mechanism to explain how this occurs without S-K entrapment. According to the first (and simplest), DNA moves 'upwards' between disengaged ATPase heads, and is subsequently trapped in the E-S compartment following ATP-driven head engagement (Figure 9A). An alternative is that a loop of DNA is inserted into an open S-K ring. If one of the loop's segments were located above the ATPase domains while the other below, then subsequent head engagement would lead to simultaneous entrapment in both E-S/E-K compartments (Figure 9B). Neither type of DNA movement involves passage through a gate created by opening the S-K ring, hence explaining the lack of S-K entrapment. Entrapment within E-S/E-K, but not S-K compartments, in the presence of Scc2 alone was also observed with relaxed (nicked) DNAs (data not shown).

## Entrapment of DNA within E-S compartments disrupts coiled coil interactions proximal to the heads

To address whether the coiled coils are associated when Smc1/3 heads engage in vitro in the presence of Scc2, we combined the E-specific cysteine pair with one specific for the coiled coils (Smc1K201C-Smc3K198C), in close proximity to the joint (*Figure 6A*). This revealed that double crosslinking can indeed occur in the presence of ATP (*Figure 6B*). As expected, double crosslinking also occurred efficiently when the coiled coil pair was combined with one within the hinge interface (Smc1K639C Smc3E570C). These cysteine pair combinations enabled us to measure entrapment within two sub-compartments within the E-S compartment (*Figure 6A* iii): one created by simultaneous crosslinking of the hinge and coiled coils (C-H compartment) (*Figure 6A* ii) and a complementary one made by the simultaneous crosslinking of coiled coils and engaged heads (E-C compartment) (*Figure 6A* i). If DNAs entrapped in E-S compartments are in molecules whose coiled coils are associated, at least in the vicinity of their joint regions, then they must be entrapped either in the E-C or the C-H sub compartments. On the other hand, if the entrapment of DNAs within E-S compartments is accompanied by (or indeed causes), dissociation of the coiled coils in the vicinity of the Smc1K201C-Smc3K198C cysteine pair, then DNA should not be trapped in either of these sub-compartments.

Despite efficient crosslinking at both cysteine pairs (*Figure 6B*), few if any DNAs were entrapped in the presence of Scc2 and ATP following BMOE treatment of cohesin trimers containing hinge and coiled coil cysteine pairs (*Figure 6C*). Likewise, few if any DNAs were entrapped by cohesin trimers containing both E-state and coiled coil cysteine pairs. We deduce from this result that entrapment of DNAs within E-S compartments in the presence of Scc2 is accompanied by dissociation of their coiled coils in a manner that precludes crosslinking between Smc1K201C and Smc3K198C. It is important to point out that this feature was not apparent when analysing the crosslinking efficiency between Smc1K201C and Smc3K198C under the same conditions (data not shown), namely in the presence of Scc2, ATP and DNA. To explain this, we suggest that despite the addition of DNA, complexes exist, at least transiently, that have engaged their heads and have zipped up their coiled coils but have not in fact trapped DNA within their E-S compartments. Consistent with this notion is our finding that simultaneous crosslinking of coiled coils and engaged heads occurs efficiently even in the absence of DNA (*Figure 6B*).

## The DNA is 'clamped' between Scc2 and the engaged heads during entrapment within E-S/E-K compartments, as revealed by cryo-EM

The dissociation of Smc1 and Smc3 coiled coils in the vicinity of their joint regions would create space for DNA to bind to engaged heads, as observed in Mre11/Rad50 complexes (*Liu et al., 2016*) and more recently in both human and *S. pombe* cohesin containing Scc2[NIPBL/Mis4] and Scc3[SA2/Psc3]

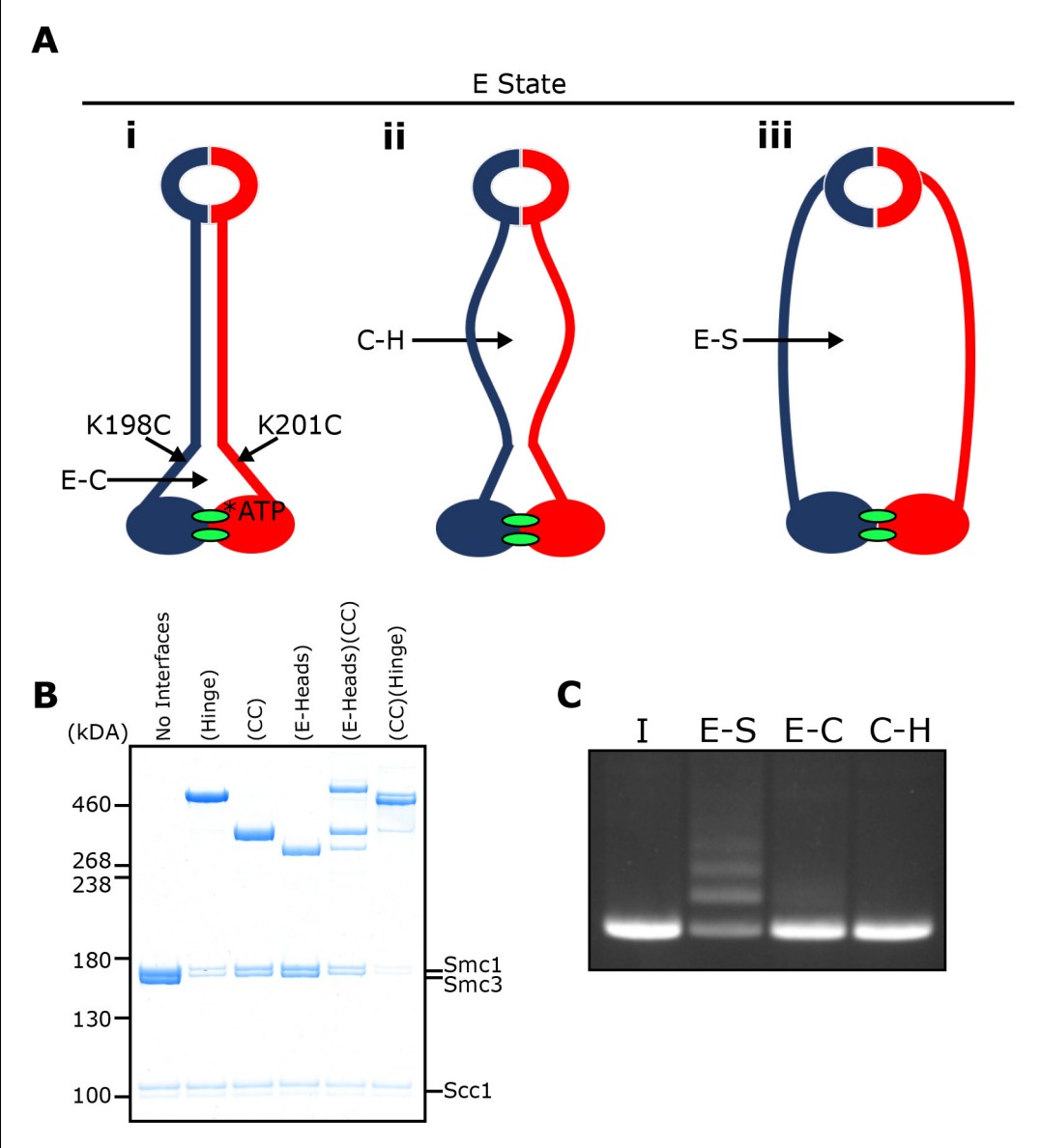

**Figure 6.** E-S/E-K entrapment leads to dissociation of the coiled coil around the joint. (**A**) Scheme showing the location of the joint cysteine pair and how head engagement could lead to different degrees of coiled coil dissociation and sub-compartment formation. (**B**) BMOE crosslinking of cohesin containing cysteine pairs at the specified interfaces in the presence of ATP. CC = coiled coils. (**C**) DNA entrapment in E-S compartments, or either E-C or C-H sub-compartments, in the presence of Scc2, incubated for 2 min (I = input DNA).

(*Higashi et al., 2020*; *Shi et al., 2020*). In the latter structures, the coiled coils of Smc1 and Smc3 diverge from each other at 70° as they emerge from the engaged heads, thereby creating a site for DNA to bind a surface on top of the heads. The DNA is also bound by Scc2, whose simultaneous association with Smc1's ATPase head and the coiled coil emerging from Smc3's head creates a new type of sub compartment within which DNA bound to the engaged heads is entrapped or 'clamped'.

Our findings that Scc2 and DNA together promote head engagement (*Figure 5—figure supplement 2D & E*) and that Scc2-4E reduces E-S/E-K entrapment (*Figure 5E*) raise the possibility that DNAs entrapped within E-S/E-K compartments through the action of Scc2 are bound in a similar manner. However, since the human (PDB 6WG3) and *S. pombe* cryo-EM complexes (*Higashi et al., 2020*; *Shi et al., 2020*) were formed by cohesin containing Scc3$^{SA2/Psc3}$, the complexes described

here, which are formed in the absence of Scc3, could in fact have a very different conformation. Furthermore, because it is not possible to trace the entire kleisin chain, and because the complexes contain only short linear DNA molecules, the topology of DNA's association with the SMC-kleisin trimers in the existing structures cannot be inferred definitively. In other words, it is not possible to determine whether the DNAs in the structures are entrapped within E-S and E-K and/or S-K compartments. As it happens, the kleisin path deduced for PDB 6WG3 (*Shi et al., 2020*) suggests that DNA, if it were circular, would be trapped in E-S and S-K but not E-K compartments. Thus, the presence of Scc3$^{SA2}$ in this complex may have had an important influence on the topology of cohesin's association with DNA.

To elucidate how DNA actually associates with EQEQ cohesin trimers in the presence of ATP and Scc2, but lacking Scc3, namely under conditions in which DNAs are clearly entrapped within E-S and E-K but not S-K compartments, we used cryo-EM to solve the structure of EQEQ cohesin trimers (Smc1, Smc3 and Scc1) bound to ATP, linear DNA (40 bp), and Scc2C2 (residues 151–1493) to a resolution of 3.4 Å (*Figure 7A*). Processing followed standard cryo-EM single particle procedures as implemented in RELION 3.1 pipelines (*Scheres, 2012*), but significant preferred orientation required the use of tilted data acquisition (Methods). With the help of previous crystal structures (PDBs 5ME3, 1W1W and 4U × 3) (*Chao et al., 2017*; *Gligoris et al., 2014*; *Haering et al., 2004*) the electron density map enabled us to build and refine a reliable atomic model, with DNA rigidly clamped between the head domains and the HAWK subunit Scc2 (*Figure 8A*, *Supplementary file 1*).

A key question arising from our crosslinking studies concerns the mechanism by which circular DNA is entrapped within E-S/E-K compartments without being entrapped within the S-K ring, namely whether a single segment of DNA is passed between the heads prior to their engagement or whether a loop of DNA is first passed through the S-K ring before head engagement traps one segment of the loop above and another below the heads. Visualising how DNA is actually grasped by cohesin and Scc2 under these conditions should in principle be revealing. However, the linearity of the 40 bp oligonucleotide used for our high-resolution structure (*Figure 7A*) precludes any conclusions as to the topology of its association with cohesin. In other words, we cannot say whether it corresponds to E-S/E-K entrapment. It also precludes any insight as to how DNA actually enters the Scc2-SMC clamp because the DNA could either have been passed through the heads prior to their engagement (or any other gate), or it could simply have been threaded through the clamp after head engagement. For this reason, we also solved to a resolution of ~7 Å the structure of the same clamped state associated with circular relaxed DNA (*Figure 7B & C*, *Supplementary file 1*). Crucially, the structure associated with circular DNA is virtually identical to that associated with the linear oligonucleotide and since the former is known to involve E-S/E-K but not S-K entrapment, we can with some certainty infer the path of the kleisin chain with respect to the DNA (*Figure 8F*). Two important conclusions can therefore be drawn. First and foremost, E-S/E-K entrapment does indeed arise from the clamping of DNA between Scc2 and engaged heads in the manner revealed by both high (*Figure 7A*) and medium (*Figure 7B*) resolution cryo-EM structures. Second, because the circular DNA is demonstrably not highly-bent when clamped (*Figure 7B*, right), the DNA must have entered the clamped state and thereby been entrapped in E-S/E-K compartments without formation of a loop (*Figure 9B*). In other words, a single segment of DNA must have been passed between the heads prior to their engagement (*Figure 9A*).

Processing of the linear DNA dataset using boxes large enough to cover entire cohesin complexes (*Supplementary file 1*, Materials and methods) revealed a significant subset of particles (~30%) that contained lower-resolution information about the conformation of the coiled coils and placement of the hinge (*Figure 7D & E*). Despite partial head-proximal unzipping of the coiled coils in this state, the reconstructed map clearly revealed a folded conformation reminiscent of *apo* condensin, a fraction of ATP-bound condensin (*Lee et al., 2020*) and *apo* cohesin (*Bürmann et al., 2019*). A folded state was also presumed to exist in the human cohesin structure where the HAWKS and the hinge interacted directly, but the coiled coils could not be resolved (*Shi et al., 2020*). Our map did not have sufficient resolution to determine whether the hinge was partially open or closed (*Figure 7E*).

## Binding of DNA to Scc2 is consistent with Scc2-4E effects

Binding of DNA by Scc2 in *S. cerevisiae* cohesin occurs in a similar manner to that recently described for human Scc2$^{NIPBL}$ (*Shi et al., 2020*; *Figure 8A*). Scc2 holds DNA by way of a curved basic and

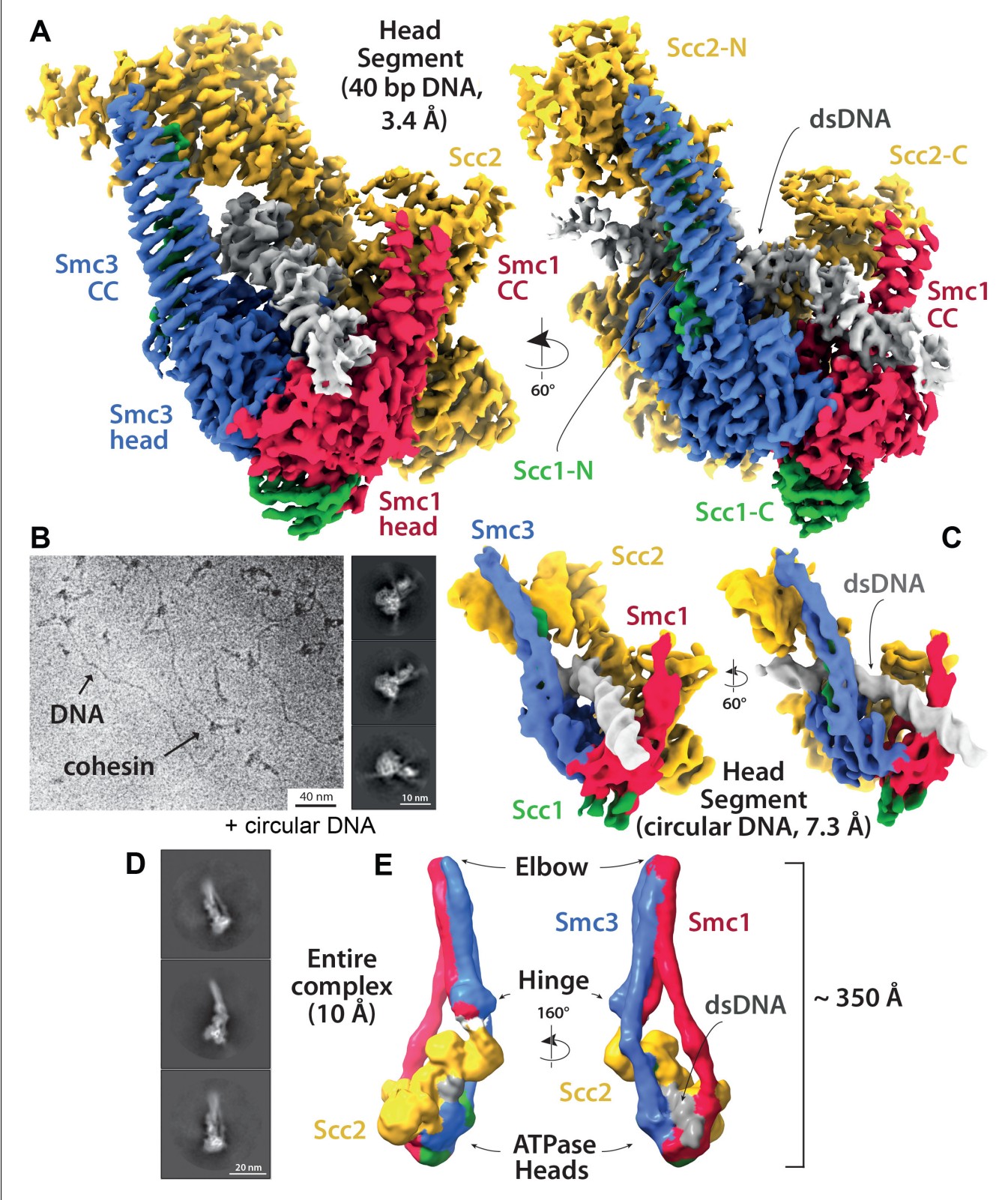

**Figure 7.** Cryo-EM of cohesin clamping DNA in the E-S/E-K state. (**A**) Cryo-EM map of 40 bp DNA clamped by Scc2- and ATP-bound cohesin EQEQ trimer at 3.4 Å resolution. Both front and side views are coloured by subunit. (**B**) Same complex as shown in A but bound to ~1.8 Kbp relaxed circular DNA as a cryo-EM field view (using Volta phase plate, left) and a selection of 2D class averages (right) clearly showing DNA emanating from cohesin/ Scc2 complexes. (**C**) 7.3 Å resolution cryo-EM map of the complex shown in B, coloured by subunit, demonstrating that the same conformation of the

*Figure 7 continued on next page*

*Figure 7 continued*

complex has been obtained as with linear DNA (panel A). Same orientations and colours as in A. (**D**) 2D class averages obtained by reprocessing of the same data set as used for A with an enlarged box size show the position of the coiled coils and the hinge. (**E**) ~ 10 Å resolution cryo-EM map of the entire tetramer complex as shown in D. Since we used the same complex as used in the in vitro entrapment reactions, we can deduce that the DNA within the clamped structure depicted in A, C and E must be entrapped in both the E-S and E-K compartments.

polar surface located around the transition between its neck and head regions (*Figure 8B*). The surface, which causes the DNA to bend slightly (~9°), is created by the spatial arrangement into a semi-circle of a series of residues (e.g. S508, N557, K714, S753, S783, and K1324/25) from the ends of six α-helices and one loop (containing K427) that together engulf the phosphate backbone. This region includes all four positions mutated in Scc2-4E (S717E K721E K788E H789E, *Figure 4*), thus neatly explaining why the charge-reversing mutations lowered the binding affinity for DNA and inhibited entrapment of DNAs in vitro.

## Scc2 binds both Smc1 and Smc3

Clamping involves not only the binding of DNA to Scc2 and to engaged heads (see below) but also entrapment in a novel compartment created by Scc2's association with both Smc1 and Smc3. The latter involves multiple binding sites (*Figure 8C*) and is therefore much more delocalized than Scc2's DNA binding. Highly prominent is the binding of Scc2 to Smc1 through the docking of HEAT repeats 18–24 (residues 1127–1493) onto the F-loop (residues 1095–1118) on Smc1 head's C-lobe and the coiled coils that emerge above (*Figure 8C* i). This mode of association is highly analogous to the binding of *S. cerevisiae* condensin Smc4 by the HAWK Ycs4 (*Lee et al., 2020*; *Figure 8—figure supplement 1*). Scc2 simultaneously interacts with Smc3's N-lobe - thereby providing a mechanism by which it promotes head engagement and subsequent ATP hydrolysis. One key contact in this regard involves residues in an otherwise disordered loop (1178–1203) that form a β-strand which docks onto the end of the central β-sheet of the Smc3 head (*Figure 8C* iii). More conserved is a major contact mediated by salt bridges between a collection of highly conserved aspartate and glutamate residues located within two loops of Scc2 (819-EDEED-823 and 781-DD-782) and two key lysine residues (K112 K113) in Smc3 (*Figure 8C* ii). Entrapment of DNA between Scc2 and engaged heads arises because in addition to the above contacts, Scc2 contacts Smc3 through HEAT repeats 1–4 (residues 151–409), which bind to the start of the joint module (coiled coil arm, residues 999–1004) in a manner that — when compared to the unbound crystal structure (*Kikuchi et al., 2016*) — causes a conformational rearrangement of Scc2's head segment. This movement is necessary to accommodate the simultaneous binding of DNA by both Scc2 and the heads while presumably playing a role in stabilising the unzipped conformation of the coiled coils (*Figure 8C* iv). Several residues within this interface (for example, Smc3 R225 K228 and Scc2 E304) are highly conserved, suggesting that it has an important function.

## Pseudo-symmetric binding of DNA to the engaged Smc1/3 ATPase heads

While Scc2 holds the upper half of the DNA's backbone through a spiral of basic and polar residues (*Figure 8B*), the engaged heads of Smc1 and Smc3 produce a 2-fold pseudo-symmetrical ABC ATPase heterodimer that binds DNA through two sites that are exactly two turns of the DNA apart and coincide with the major groove in the DNA (*Figure 8D*). As expected for DNA-binding proteins that are not sequence specific, neither protein inserts residues into the major or minor grooves and both rely solely on interactions with the DNA backbone. A consequence of this binding mode is that the two-fold symmetry of DNA is matched almost perfectly by the Smc1/3 heterodimer (*Figure 8D* top). The two pseudo-symmetrical DNA binding sites close to the major groove are formed through basic and polar amino acids in Smc1 (e.g. K63, S112, R113, and K124) and Smc3 (e.g. K57, K112, K113, and K125) (*Figure 8D* bottom). The DNA is bent slightly (~9°) and it seems likely that without bending the DNA binding sites on Smc1 and Smc3 would be too close together. Overall, DNA binding is linked to head engagement and the ATPase cycle, as the complete binding path for DNA along the heads only arises when both heads come together in the E-state.

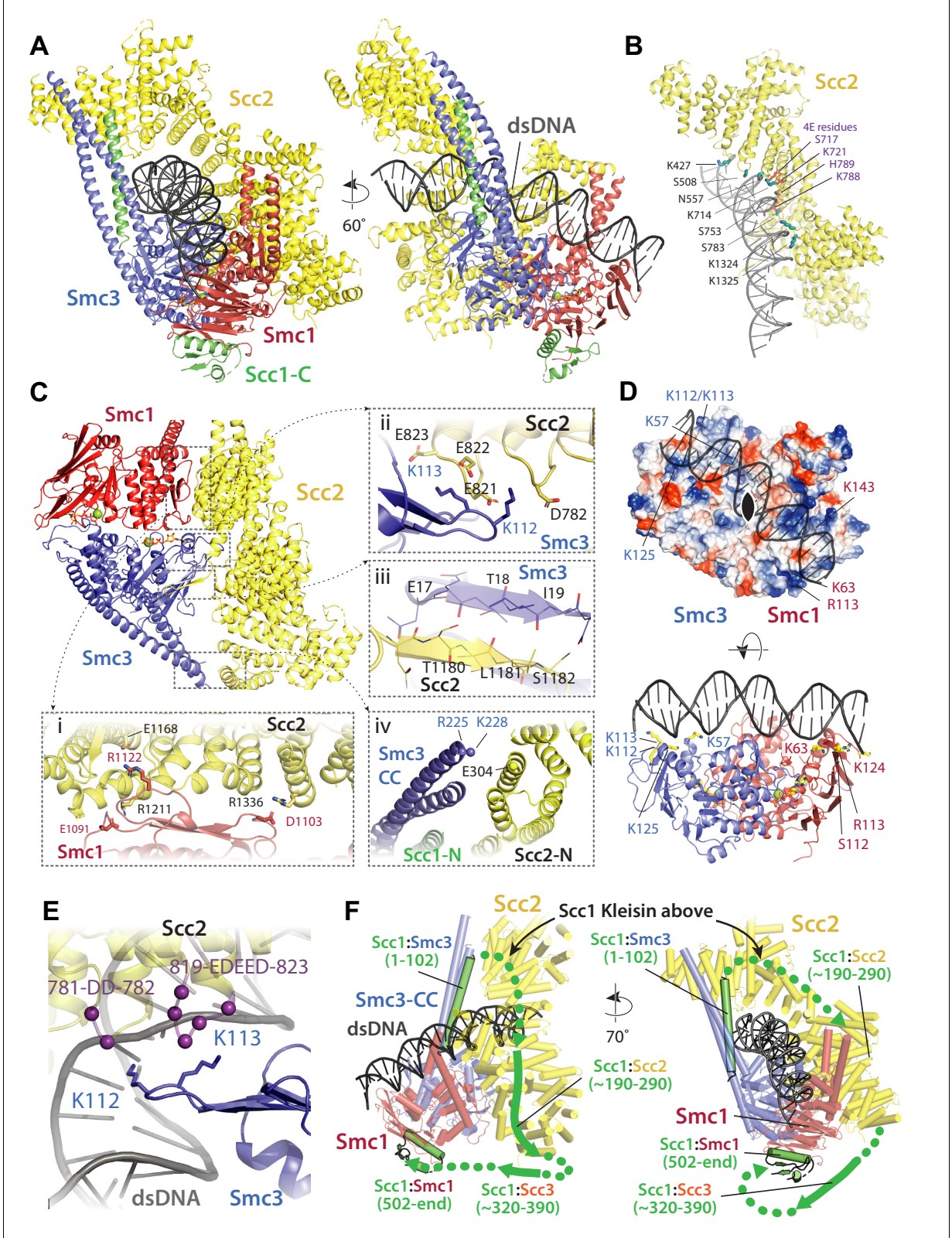

**Figure 8.** Molecular interactions in the E-S/E-K state. (**A**) Cartoon representation of the refined atomic model of cohesin's clamped (E-S/E-K) state based on the 3.4 Å resolution cryo-EM map (*Figure 7A*, same orientation and colours, *Supplementary file 1*). (**B**) Basic and polar residues of Scc2 involved in the interaction with DNA. Scc2 interacts only with the backbone. Residues in its vicinity are labelled in black while those mutated in Scc2-4E (*Figure 4*) in purple. (**C**) Scc2 makes extensive contacts with both Smc1 and Smc3 heads: (i) Scc2 binds Smc1 through its HEAT repeats 18–24 (residues

*Figure 8 continued on next page*

**Figure 8 continued**

1127–1493) that dock onto the F-loop on Smc1 (residues 1095–1118) and the emerging coiled coils above it. (ii) Smc3's K112 K113, whose acetylation reduces loading efficiency, are in the vicinity of a negatively charged patch on Scc2 (819-EDEED-823 and 781-DD-782). (iii) Scc2 binds to Smc3 through a β-strand (part of the otherwise disordered loop 1178–1203) that complements the central β-sheet of Smc3. (iv) The N-terminal section of Scc2 contacts parts of Smc3's coiled coil arm/neck, close to where the last ordered region of Scc1's N-terminal domain is bound to the Smc3 coiled coil. (D) DNA binding to the SMC head domains is pseudo-symmetrical. Top: the pseudo two-fold axis of the DNA neatly aligns with that of the head domains underneath. Bottom: The head domains interact with the DNA almost exactly two full DNA turns apart, utilising pseudo symmetry-related surfaces (Smc1: K63, S112, R113, and K124; Smc3: K57, K112, K113, and K125). (E) The two lysines K112 K113 are in contact with a negatively charged patch on Scc2 (see panel C iii), but are also in the vicinity of the DNA backbone. (F) The N-and C-terminal domains of the kleisin Scc1 bind canonically to Smc3 and Smc1, linking the heads and topologically closing the tripartite Smc1/Smc3/Scc1 (S–K) cohesin ring. A tentative path of the disordered regions of Scc1, not visible in our cryo-EM map is shown to demonstrate the topology as deduced from the loading reactions and subsequent crosslinking that show that the DNA must be outside the tripartite S-K ring.

The online version of this article includes the following figure supplement(s) for figure 8:

**Figure supplement 1.** Comparison of cryo-EM structures.

## How do Smc3 K112 and K113 affect loading?

In yeast, Smc3 K112 and K113 have important roles in loading of cohesin onto chromosomes. We show here that changing KK to QQ reduces S-K entrapment in the presence of Scc2 and Scc3 in vitro (*Figure 1—figure supplement 1C*), recapitulating the adverse effect on genome wide association in vivo (*Hu et al., 2015*). The QQ double mutation is thought to mimic acetylation of K112 K113, which takes place as cells undergo S phase and may have a role in altering how cohesin interacts with DNA, principally whether it can associate (de novo) with and translocate along chromosomes. Our high resolution cryo-EM structure (*Figure 8A and C*) reveals that K112 K113 belong to the array of residues that create a basic environment for charge-mediated binding to the DNA backbone (*Figure 8E*). This raises the possibility that K112 K113 participate directly in the binding of DNA to engaged heads. However, close inspection of their side chains shows that they in fact face towards the two aspartate and glutamate rich loops in Scc2 (819-EDEED-823 and 781-DD-782) (*Figure 8C* ii), implying that they engage in ionic interactions between Smc3 and Scc2 as well as or instead of DNA. If so, one consequence of acetylation or replacement by QQ may be disruption of this mode of Scc2-Smc3 binding, a notion consistent with our previous finding that QQ greatly reduces stimulation of cohesin's ATPase activity by Scc2 in the absence of DNA, at least when Scc3 is present (*Petela et al., 2018*).

Remarkably, the charge reversal substitution Scc2E822K was isolated as a spontaneous mutation that suppresses the lethality of *scc4* mutants whose Scc2's activity is greatly compromised (*Petela et al., 2018*). Because Scc2E822K would be predicted to reduce binding to Smc3 K112 K113, which might have been expected to further reduce, not improve, the compromised Scc2 activity of these *scc4* mutants, we suggest that E822K might loosen, but not eliminate, the ionic interactions between Scc2 and K112 K113. This may enable K112 K113 to make a greater contribution to DNA binding and thereby increase the affinity between engaged heads and DNA. Given the extreme conservation of residues equivalent to Smc3 K112 K113 and Scc2 E822 D823, it seems likely that the interface has a similar function in most eukaryotes and yet mutations equivalent to *smc3 K112Q K113Q* in *S. pombe* and in human tissue culture cells (where they are not lethal) (*Feytout et al., 2011*; *Ladurner et al., 2016*) do not eliminate cohesin loading in the manner observed in yeast (*Davidson et al., 2016*). Cohesin's ATPase is necessary for LE as well as for loading and we therefore suggest that QQ mutations may turn out to compromise LE.

Smc3 K112 K113 are required for Wapl-dependent release of cohesin from chromosomes as well as for optimal ATPase activity (*Ladurner et al., 2014*; *Petela et al., 2018*). Because, release only occurs when Scc2 is replaced by Pds5, our cryo-EM structures provide little direct insight as to their role during release. A key question is whether K112 K113 interact with Pds5 in a similar manner to Scc2 or whether their primary role during release is to bind DNA. Acetylation during S phase blocks release and helps to stabilize Pds5's association with chromosomal cohesin.

## Scc1 is bound to both heads and does not engage in DNA binding

Though our map (*Figure 7A*) shows little to no density for residues of Scc1 known to bind the central cleft of Scc2 (Scc1 residues ~ 190–290), it shows very clearly that Scc1's two structured domains,

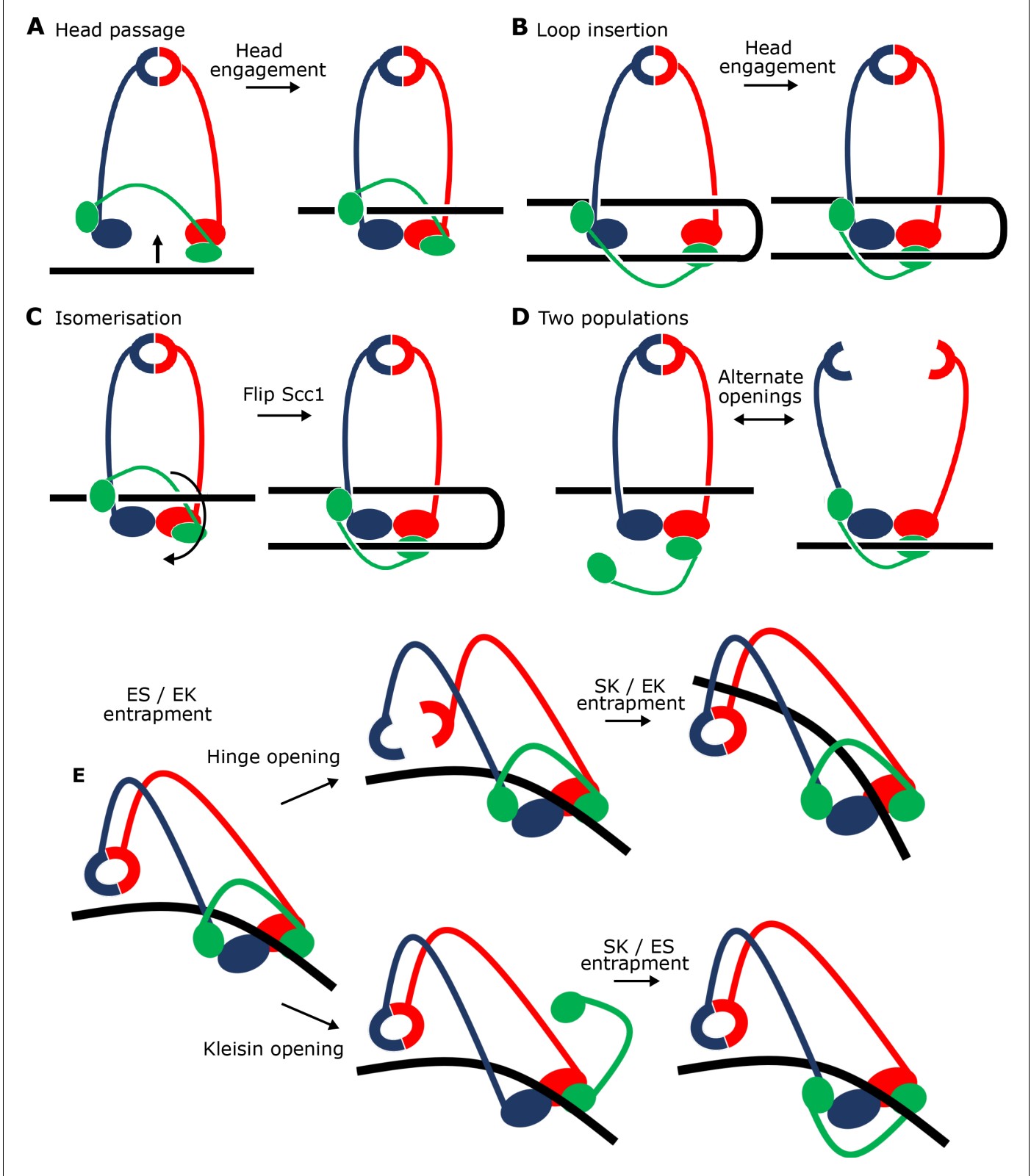

**Figure 9.** Potential mechanisms for Scc2-driven E-S/E-K entrapment and subsequent S-K ring entrapment. (**A**) ES/EK entrapment by DNA passing through open heads, or (**B**) through a DNA loop being inserted. (**C**) Topological isomerism between A and B. (**D**) ES/EK entrapment due to two distinct populations. (**E**) Models for converting E-S/E-K entrapment to S-K entrapment.

Scc1-N (residues 67–103) and Scc1-C (residues 502–555), are bound in a canonical manner to Smc3's neck and the base of Smc1's ATPase respectively (*Gligoris et al., 2014*; *Haering et al., 2004*) thereby bridging the two heads to form the S-K ring (*Figure 8F*). We can therefore exclude the possibility that head engagement, at least in the presence of Scc2 and DNA, causes the sort of rearrangement of Smc3's coiled coil thought to induce Scc1-N's release from Smc3, as suggested by recent structural studies of an ATPγS-bound cohesin trimer (*Muir et al., 2020*). Similarly, even though it is evident that there is a high level of conservation between the structures of human (*Shi et al., 2020*), *S. pombe* (*Higashi et al., 2020*) and *S. cerevisiae* cohesin reported here, the latter shows no sign of any structured part of Scc1 participating in the binding of DNA. Indeed, despite its conservation among most eukaryotes, the positively charged loop within Scc1-N that binds DNA in the human and *S. pombe* complexes (residues 23–28 in both) is not present in yeast.

## Scc2 and DNA disrupt the J-state in the absence of ATP

Passage of DNA between disengaged heads prior to ATP-driven head engagement would be essential for entrapment simultaneously in E-S and E-K compartments (see Discussion). There must therefore exist a mechanism by which the heads are moved sufficiently far apart to permit DNA passage. To investigate this, we tested the effect of ATP, Scc2, and DNA on crosslinking between the J-state cysteine pair. Conditions that promote efficient E-state crosslinking, namely addition of ATP, Scc2, and DNA, caused a modest ~20% reduction in J-state crosslinking (*Figure 5—figure supplement 2F*), confirming that the E-state is formed at the expense of J. As expected, other combinations of these three factors had less effect. Surprisingly, addition of Scc2 and DNA in the absence of ATP had the greatest effect, causing a ~ 50% reduction in J crosslinking, an effect that was highly reproducible. Such a marked reduction is presumably caused by the heads adopting a different conformation. Importantly, this does not correspond to the E-state as very little crosslinking takes place between Smc1N1192C and Smc3R1222C under these conditions (*Figure 5—figure supplement 2A*). We therefore suggest that in the absence of ATP, both Scc2 and DNA reduce J-specific crosslinking by driving or indeed holding the ATPase heads apart, a process that could facilitate passage of DNA between them and thereby facilitate its entrapment in E-S compartments when heads engage in the presence of ATP.

# Discussion

## In vitro reproduction of DNA entrapment within cohesin SMC-kleisin rings

In vivo studies have shown that cohesin entraps circular minichromosomes within its S-K ring (*Gligoris et al., 2014*; *Srinivasan et al., 2018*). We demonstrate here that purified cohesin possesses such an activity also in vitro. Unlike previous assays that have merely measured the physical association between cohesin and DNA and investigated its resistance to salt or sensitivity to kleisin cleavage (*Murayama and Uhlmann, 2015*; *Murayama and Uhlmann, 2014*), our method measures topological association directly. By covalently circularising the cohesin ring and its component compartments we can make unambiguous deductions about the topology between DNA and cohesin. The entrapment of DNAs within S-K rings measured by this method depends on Scc2, Scc3, and ATP. Importantly, it is also stimulated by ATP hydrolysis, a feature that has been lacking in previous assays but is of paramount importance for entrapment in vivo (*Srinivasan et al., 2018*). Four other key properties of the in vitro S-K entrapment activity reflect cohesin's behaviour in vivo, namely it depends on the ability of Scc2 and Scc3 to bind DNA, on the ability of ATP to bind Smc3 heads, and on Smc3's K112 K113 residues, whose lack of acetylation is necessary for loading in yeast (*Hu et al., 2015*). We therefore suggest that the in vitro DNA S-K entrapment described here involves mechanisms similar or identical to those of cohesin operating within cells. If we assume that the efficiency of BMOE induced S-K circulation is around 20%, we estimate that many DNAs are entrapped by 15 or more cohesin rings in our assay after a 40 min incubation.

## Potential mechanisms for Scc2-driven E-S/E-K entrapment

An obvious question concerns the state of the Smc1 and Smc3 ATPase heads when DNA is entrapped. They could either be engaged in the presence of ATP, juxtaposed together in the

absence of ATP (a state facilitated by extensive association of the Smc1 and Smc3 coiled coils), or fully disengaged. To address this, we used a Smc1-Smc3 cysteine pair specific for engaged heads (Smc1N1192C Smc3 R1222C), which revealed that DNAs are also entrapped efficiently between the hinge and engaged heads (the E-S compartment), between engaged heads and the kleisin subunit associated with them (the E-K compartment), but only rarely between juxtaposed heads and their associated kleisin (the J-K compartment). For obvious reasons, we were not able to address using cysteine-specific crosslinking whether DNAs are also entrapped within S-K rings with fully disengaged heads. However, given that cohesin's ability to hydrolyse ATP is important for S-K entrapment, it is likely that at least some S-K rings that have entrapped DNA in vitro are in this state. Our failure to observe efficient entrapment within J-K compartments was unexpected given that this state has been documented in vivo (*Chapard et al., 2019*).

We were surprised to find that unlike entrapment within S-K rings, which requires Scc3, entrapment within E-S/E-K compartments was entirely Scc3 independent. In other words, entrapment within E-S/E-K compartments in the presence of Scc2 alone is not accompanied by entrapment within S-K rings. The similarity in kinetics suggests that entrapment within E-S and E-K compartments driven solely by Scc2 occurs simultaneously as part of the same reaction. The simplest explanation for this is that DNA is transported (upwards) between disengaged ATPase heads and then subsequently trapped in the E-S compartment due to ATP-driven head engagement (*Figure 9A*). The simultaneous entrapment within E-K compartments arises naturally from this, as the kleisin polypeptide must be looped 'upwards' to accommodate DNA entry in this manner. Crucially, this process would not require opening of the hinge or either SMC-kleisin interface, processes that would be necessary for entrapment within S-K rings. Our cryo-EM structures reveal that entrapment within E-S/E-K compartments is accompanied and probably driven by the binding of DNA to Scc2 and DNA binding sites on the upper surface of Smc1 and Smc3 ATPase heads created upon head engagement. In other words, entrapment within E-S/E-K compartments (in the absence of S-K entrapment) arises from the clamping of DNA between Scc2 and engaged heads. The remarkable similarity between this structure and one formed between DNA, Scc2$^{NIPBL}$, and tetrameric human cohesin (PDB 6WG3) (*Shi et al., 2020*) shows that the clamping of DNA between Scc2$^{NIPBL}$ and engaged heads not only does not involve Scc3$^{SA2}$ (whose ortholog SA2 was present in the human structure) but more importantly the same clamping happens even when Scc3 is absent (*Figure 8—figure supplement 1*).

An alternative is that simultaneous E-S/E-K entrapment arises from insertion of a loop of DNA into an open S-K ring. If one segment of this loop were located above the heads while the other located below them, head engagement would also lead to simultaneous entrapment in E-S and E-K compartments (*Figure 9B*). This scenario, which could also involve the clamping of DNA between Scc2 and engaged heads, is not only more complex, but it must somehow explain how DNA is bent prior to insertion, a process which would carry a clear entropic penalty and more importantly is not apparent from cryo-EM images of EQEQ cohesin associated with circular DNAs (*Figure 7B* right). Importantly, the states created by the two mechanisms are topologically isomeric. In other words, it is possible to transform the state described in *Figure 9A* to that of *Figure 9B* merely by moving Scc1's central domain 'downwards' to below the heads and simultaneously bending the DNA through 180°.

It is nevertheless important to point out that because our assays measuring E-S/E-K entrapment driven by Scc2 alone use complexes with different sets of cysteine pairs, they do not per se prove simultaneous entrapment of DNAs within both compartments. Thus, DNAs trapped in E-S/E-K compartments could in principle belong to separate populations (*Figure 9D*). According to this scenario, and because E-S/E-K entrapment is not accompanied by S-K entrapment, DNAs entrapped solely within E-S compartments would have to be held by complexes whose kleisin subunit had dissociated from one or both ATPase heads while DNAs entrapped solely within E-K compartments would have to be trapped by complexes whose hinge had opened. There are two arguments against this interpretation. First, it is very unclear why head engagement in the presence of ATP, Scc2, and DNA should be associated with two such different events. An even more compelling argument stems from our cryo-EM structures of DNA clamped by Scc2 and engaged heads. A low resolution structure reveals coiled coils folded around their elbow and a dimerised hinge associated with Smc3's coiled coil (*Figure 7E*), while a high resolution structure shows that both N- and C-terminal kleisin domains (Scc1-N and -C) are bound to Smc3's neck and the base of Smc1's ATPase respectively

(*Figure 8A*). In other words, association of DNA with engaged heads in the presence of Scc2 does not appear to be accompanied by opening of any of the S-K ring's three interfaces, ruling out the possibility that E-S and E-K entrapment are independent processes.

The notion that DNA can be engaged by cohesin rings in a manner that does not require opening of the hinge or either SMC-kleisin interface (*Figure 9A & B*) is consistent with the recent observation that sealing all three interfaces does not adversely affect DNA loop extrusion by human cohesin complexes (*Davidson et al., 2019*) as well as the finding that the association between chromosomes and cohesin complexes with certain hinge mutations is not accompanied by S-K entrapment in vivo (*Srinivasan et al., 2018*).

Though the path of the kleisin chain connecting the ATPase heads is not discernible in any of the cryo-EM structures (this work) (*Higashi et al., 2020*; *Shi et al., 2020*), our knowledge that circular DNAs are entrapped within E-S/E-K but not S-K compartments under identical conditions makes clear that the kleisin chain, whose Scc1-N domain is bound to Smc3's neck, must pass over the DNA bound to the engaged heads before its Scc1-C domain binds to the base of Smc1's ATPase (*Figure 9A* and molecular equivalent *Figure 8F*). This topology is not merely of academic interest as it provides crucial insight into the pathway by which DNAs are clamped by Scc2 and engaged ATPase heads. Because EQEQ cohesin's association with Scc2 and circular DNA does not appear to cause much DNA bending (*Figure 7B* right), we favour the notion that E-S/E-K entrapment arises when DNAs pass (in an upwards direction) between heads prior to their engagement in the presence of ATP and not by insertion of a loop into the S-K ring.

Given that the ATPase heads are frequently associated either in the E- or J-state, there must exist a mechanism to create an opening between them, if only transiently, in order for DNA to pass through before being clamped by their subsequent engagement. Our observation that Scc2 and DNA disrupts the J-state, albeit only in the absence of ATP, may be relevant in this regard (*Figure 5—figure supplement 2F*). This J-state disruption was not caused by adoption of the E-state (*Figure 5—figure supplement 2A*), which requires ATP, and it must therefore involve transition to a state in which Smc1 and Smc3 ATPase heads adopt yet another conformation. This could be a state in which Scc2 and DNA together drive apart the ATPase head domains, thereby enabling DNA to pass between them. Subsequent ATP binding would then cause head engagement and E-state formation, trapping DNA inside both the E-S and E-K compartments. Interestingly, a conformation of this nature has recently been observed in condensin bound to Ysc4 (*Lee et al., 2020*), where the latter bridges the Smc2 and Smc4 heads, holding them apart by some distance. Although DNA was absent from this structure, the separation of the heads would be sufficient for DNA to

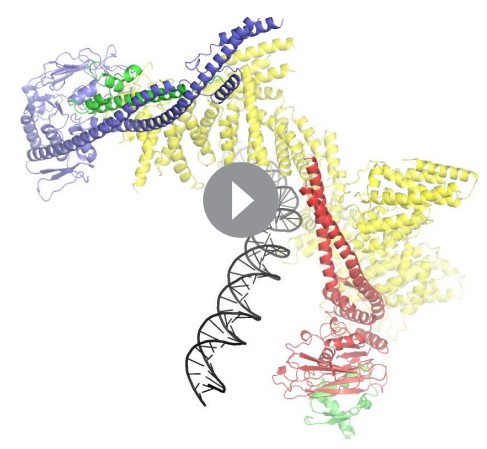

**Video 1.** A model for the formation of the clamped E-S/E-K state of cohesin. According to this model cohesin transitions from a putative 'bridged state' (modelled on the same state of yeast apo condensin as observed by cryo-EM) (*Lee et al., 2020*) in which Scc2, analogous to Ysc4 in condensin, bridges the Smc1/3 heads. In the bridged state Scc2's DNA binding surface becomes accessible for DNA to attach without impediment and positions it for the next step, namely clamping. ATP-binding driven head engagement, achieved through a rotation of the Smc3 head relative to the rest of the complex, and with Scc2's and Smc1's relative orientations staying the same, results in entrapment in the E-S compartment. Because the disordered kleisin chain has to be pushed upwards during the clamping, the DNA is also in the E-K compartment. The initial binding of DNA to the Scc2 DNA binding site guides the DNA through the large opening of the heads generated by the bridged state and leads to the final clamped state that has been described in this study. The video is a simple morph between a putative bridged state of cohesin modelled on the same state in condensin and the high-resolution cryo-EM structure of the clamped state determined in this study, with a few clashes removed manually because cohesin and condensin subunits, in particular Scc2 and Ycs4, are not completely homologous structurally.

https://elifesciences.org/articles/59560#video1

pass between them and exposes the HAWK's DNA-binding surface without any impediment. If we assume that Scc2 bridges Smc1 and Smc3 heads in a similar fashion prior to their engagement, then one merely has to propose that DNA initially binds to Scc2 while in the bridged state and remains associated as the Smc3 head pivots around and the Smc3 ATPase head engages with that of Smc1 (*Video 1*). We envisage that Scc2's association with Smc1 heads (which strongly resembles that between Ycs4 and Smc4, *Figure 8—figure supplement 1*) remains unaltered during this transition, as it did between Smc4 and Ycs4 in condensin (*Lee et al., 2020*).

Our demonstration that DNAs transported into the sub-compartment created by Scc2's association with engaged ATPase heads results in entrapment in E-S/E-K but not S-K compartments is difficult to reconcile with the proposal that DNAs must first pass through a transiently opened Scc1-Smc3 interface before they enter the clamped state created by head engagement (*Higashi et al., 2020*), a process that has been termed DNA 'gripping'. Passage through a gate created by opening the Scc1-Smc3 interface before being clamped by engaged heads and Scc2 would be accompanied by E-K and S-K entrapment but not by E-S entrapment, which is contrary to what we observe.

It is also worth pointing out that E-K entrapment would not be possible if Scc1's NTD were dissociated from Smc3's neck upon head engagement, as has been suggested by a cryo-EM structure of heads engaged in the absence of both DNA and Scc2 (*Muir et al., 2020*). The fact that E-K entrapment accompanies E-S entrapment during our Scc2-only reaction implies that Scc1's NTD does not in fact dissociate from Smc3's neck upon head engagement when DNA and Scc2 are present, a feature also revealed by cryo-EM of our yeast (*Figure 8A*), human (*Shi et al., 2020*) and *S. pombe* (*Higashi et al., 2020*) structures of cohesin heads in complex with DNA and Scc2[NIPBL/Mis4]. Because Scc2 is necessary to prevent cohesin's release from chromosomes during G1, and because release is accompanied by disengagement of Scc1's NTD from Smc3's neck (*Srinivasan et al., 2019*), we suggest that head engagement may indeed promote Scc1's dissociation from Smc3 but that this process is actively inhibited by Scc2. One of the functions of Pds5 and Wapl in mediating release during G1 when Smc3 is not acetylated may be to replace Scc2 and thereby abrogate this protection mechanism. In this regard, it is interesting that Scc2 contacts the joint region within Smc3's coiled coil adjacent to where Scc1's NTD binds to Smc3's neck (*Figure 8C* iv), an interaction also observed in the human structure (*Shi et al., 2020*) and could have a role in hindering Scc1's dissociation from Smc3 upon head engagement.

The remarkable similarity in the structures by which yeast and human cohesin clamp DNA between Scc2[NIPBL] and engaged ATPase heads (*Figure 8—figure supplement 1*) suggests that this highly conserved conformation must have crucial physiological functions. We propose two possibilities. The first is as follows. Because Scc2 is required for S-K entrapment in the presence of Scc3, as well as for E-S/E-K entrapment in its absence, we suggest that entrapment driven by Scc2 and the binding of DNA to engaged heads is necessary for subsequent S-K entrapment. However, this does not exclude the possibility that Scc2 has roles in S-K entrapment additional to formation of an E-S/E-K intermediate. In other words, the clamping of DNAs between Scc2[NIPBL] and engaged heads may be a key intermediate during the process of S-K entrapment and hence crucial for the establishment of sister chromatid cohesion.

The observation that S-K entrapment is clearly not necessary for DNA translocation or loop extrusion and may in fact be a rare event in the life of chromosomal cohesin suggests another possibility. The clamping of DNA between Scc2[NIPBL] and engaged heads and subsequent release upon ATP hydrolysis, all in the absence of S-K entrapment, may be the driving force for cohesin's translocation along DNA, a notion fully consistent with Scc2's key role in stimulating DNA-dependent ATP hydrolysis (*Petela et al., 2018*) and loop extrusion (*Davidson et al., 2019*). If so, a crucial question for the future is how transport of DNA into the sub-compartment created by Scc2 and engaged heads is harnessed to mediate translocation along DNA. We presume that DNA translocation is accompanied (and indeed driven) by recurrent cycles of DNA uptake into the clamped state, with each cycle involving segments of DNA further along the chromosome fibre. However, functional translocation would not be possible without a second (reciprocal) mechanism by which DNAs are recurrently bound and released. Scc3's ability to bind DNA may be crucial in this regard. Another idea is that cohesin's hinge provides the second site and that the clamp/release transport cycle is accompanied by changes in the folding of Smc1/3 coiled coils around their elbow region, which could be the key to walking along the DNA. However, this notion is difficult to reconcile with the observation that

cohesin's coiled coils can be folded whether its ATPase heads are engaged (*Figure 7E*) or disengaged (*Bürmann et al., 2019*).

## Scc3 catalyses entry of DNA inside the SMC-kleisin ring

Our finding that Scc3 is essential for S-K but not E-S/E-K entrapment reveals that Scc3 has a unique role in promoting entry of DNA inside the SMC-kleisin ring as well as being necessary for loop extrusion (*Davidson et al., 2019*). In principle, Scc3 could catalyse DNA entry either via a gate created by transient hinge opening or through one produced by transient dissociation of one or both SMC-kleisin interfaces (*Figure 9E*). Crucially, S-K entrapment in vivo is not abolished by fusing Scc1's NTD to Smc3 or by fusing its CTD to Smc1, implying that DNA must enter either through the hinge or through dissociation of either one of the two SMC-kleisin interfaces (possibly through simultaneous dissociation of both) (*Srinivasan et al., 2018*). There is little or no direct evidence regarding which mechanism is correct. SMC-kleisin dissociation has been strongly implicated in release and is therefore also a plausible mechanism for entry (*Beckouët et al., 2016*). Nevertheless, hinge opening is equally plausible, especially in the light of recent findings that folding of cohesin's coiled coils around an elbow brings its hinge domain into close proximity to DNA bound to the heads, and that Scc3$^{SA2}$ interacts with a half opened hinge when DNA is bound to human cohesin-Scc2$^{NIPBL}$ complexes (*Bürmann et al., 2019*; *Shi et al., 2020*). Ascertaining which mechanism is at play will require a method to measure the effect on S-K entrapment of chemically linking interfaces together in a manner that is orthogonal to the BMOE-induced crosslinking. For example, prior crosslinking of both SMC-kleisin interfaces would abolish entrapment via a kleisin gate (*Figure 9E* bottom pathway) but not via a hinge gate (*Figure 9E* top pathway).

This feature of Scc3's activity depends on its ability to bind DNA in a manner similar to that employed by Scc2 (this work) (*Li et al., 2018*; *Shi et al., 2020*) and condensin's Ycg1 HAWK (*Kschonsak et al., 2017*). Two sets of residues are implicated in DNA binding (K224 K225 R226 and K423 K513 K520). Mutation of one or other set does not abrogate DNA binding or cause lethality but does reduce cohesin's association with chromosomes while mutation of both sets (Scc3-6E) abolishes not only DNA binding and S-K entrapment in vitro but is lethal and abolishes all loading throughout the genome in vivo. Though it abrogates entrapment of DNA within S-K compartments (*Figure 3F*), Scc3-6E has no effect on Scc2 driven E-S/E-K entrapment (data not shown). Thus, if S-K entrapment in vivo involved prior formation of an E-S/E-K intermediate, which is consistent with the latter's more rapid kinetics in vitro, then cohesin containing Scc3-6E should form this intermediate and accumulate in this state, possibly at loading sites. Our observation that, despite failing to associate with the vast majority of the genome, Scc3-6E cohesin accumulates at especially high levels at *CEN* sequences, which are highly efficient loading sites, suggests that this may indeed be the case. Unlike Scc3-6E, complete depletion of Scc3 abrogates cohesin's association at *CEN*s as well as along chromosomes arms, which implies that Scc3 has additional functions that do not involve or require its ability to bind DNA.

The notion that entrapment of DNA within S-K rings is preceded by its prior entrapment within E-K/E-S compartments by Scc2 and engaged SMC heads in a manner observed in our cryo-EM structure raises the interesting possibility that DNA is eventually entrapped within the S-K ring, not by passing from outside to inside, but instead by being allowed to exit from either the E-S or the E-K compartment by transiently opening one of the S-K ring's three interfaces. Transient hinge opening would permit DNA's escape from the E-S compartment (*Figure 9E* top pathway) while transient dissociation of one or another, or indeed both, kleisin-head interfaces would permit escape from the E-K compartment (*Figure 9E* bottom pathway). In both cases, the subsequent closing of these exit gates would lead to entrapment of DNA within the S-K ring. According to these scenarios, exit via the hinge or via a SMC-kleisin interface without head disengagement would lead, at least initially, to the selective loss of E-S and E-K entrapment respectively. If true, clamping of the DNA would provide the opportunity to open gates without losing grip of the DNA while doing so. It is interesting in this regard that whereas E-K entrapment does not increase between 2 and 40 min when both Scc2 and Scc3 are present, E-S entrapment continues to increase in parallel with the rise in S-K entrapment. Whether this asymmetry is a hint that Scc3 promotes entrapment within S-K rings by opening an SMC-kleisin interface will require far more rigorous types of experiments. Though EQEQ mutants reduce S-K entrapment, they do not eliminate it, suggesting that DNA entry can in principle occur without head disengagement, as depicted in *Figure 9E*.

The notion that a key function of Scc3, dependent on its ability to bind DNA, is to facilitate entrapment of DNA within S-K rings has an important corollary. S-K entrapment is thought to be a crucial feature of sister chromatid cohesion. Hitherto, direct evidence for this mechanism has been confined to the observation of small circular minichromosomes entrapped within S-K rings in vivo. We show here that a function of Scc3, not shared by Scc2, is to facilitate entrapment within S-K rings. If this is also an essential function of Scc3 in vivo, it follows that S-K entrapment must also be an essential cohesin function and one that applies to proper chromosomes as well as small circular ones.

We have known for two decades that Scc2 and Scc3 have different roles in promoting cohesin's association with chromosomes (*Ciosk et al., 2000*; *Tóth et al., 1999*). The various topological assays described in this paper have finally revealed some of these. Scc2 promotes entrapment of DNA in E-S/E-K compartments by promoting its binding to engaged SMC ATPase heads, while Scc3 promotes entrapment inside S-K rings. These findings are supported by our cryo-EM structure that reveals the molecular basis of the clamped E-S/E-K state and also by recent cryo-EM structures containing cohesin, DNA, and Scc2$^{NIPBL/Mis4}$ as well as Scc3$^{SA2/Psc3}$ (*Higashi et al., 2020*; *Shi et al., 2020*). Crucially, our assays reveal the topology of DNA's association with cohesin and the path of the kleisin for the clamped E-S/E-K state, at least when formed by Scc2 alone and head engagement.

Entrapment of DNAs within E-S compartments has not hitherto been detected in vivo, emphasizing the value of in vitro systems in revealing reactions that are otherwise difficult to detect. Future work will be required to address whether E-S/E-K entrapment also occurs inside cells, to elucidate the mechanism of S-K entrapment, and to reveal conditions that promote J-K entrapment, a form that has been detected in vivo but not yet efficiently in vitro.

# Materials and methods

### Key resources table

| Reagent type (species) or resource | Designation | Source or reference | Identifiers | Additional information |
|---|---|---|---|---|
| Strain, strain background (*S. cerevisiae*) | *MATa ura::ADH1 promoter-OsTIR1-9myc::URA3 Scc3-PK3-aid::KanMX4 SCC1-HA3::HIS3* | This study | KN20783 | |
| Strain, strain background (*S. cerevisiae*) | *MATa Scc3-PK3-aid::KanMX4 SCC1-HA3::HIS3* | This study | KN20785 | |
| Strain, strain background (*S. cerevisiae*) | *MATa/alpha scc3::NatMX4/WT* | This study | KN21079 | |
| Strain, strain background (*S. cerevisiae*) | *MATa/alpha scc3::NatMX4/WT, leu::Scc3-HA3::LEU* | This study | KN21273 | |
| Strain, strain background (*S. cerevisiae*) | *MATa Scc1-PK9::KanMX scc2-45::natMX (L545P D575G)* | This study | KN22390 | |
| Strain, strain background (*C. glabrata*) | *MATa, SCC1-PK9::NATMX4* | *Petela et al., 2018* | KN23308 | |
| Strain, strain background (*S. cerevisiae*) | *MATa Scc1-PK9::KanMX scc2-45::natMX (L545P D575G) lys2::Scc2-HyGMX* | This study | KN24185 | |
| Strain, strain background (*C. glabrata*) | *MATa, SCC1-HA3::NATMX4* | *Petela et al., 2018* | KN25532 | |

*Continued on next page*

*Continued*

| Reagent type (species) or resource | Designation | Source or reference | Identifiers | Additional information |
|---|---|---|---|---|
| Strain, strain background (*S. cerevisiae*) | *MATa Scc1-PK9::KanMX scc2-45::natMX (L545P D575G) LYS2::Scc2(S717L, K721E)-HygMX* | This study | KN27010 | |
| Strain, strain background (*S. cerevisiae*) | *MATa/alpha scc3::NatMX4/WT, leu::Scc3 (K224E, K225E, R226E)-HA3::LEU* | This study | KN27539 | |
| Strain, strain background (*S. cerevisiae*) | *MATa scc3::NatMX4, Scc1-PK6::TRP1, leu::Scc3-HA3::LEU* | This study | KN27542 | |
| Strain, strain background (*S. cerevisiae*) | *MAT alpha scc3::NatMX4, Scc1-PK6::TRP1, leu::Scc3 (K224E, K225E, R226E)-HA3::LEU* | This study | KN27547 | |
| Strain, strain background (*S. cerevisiae*) | *MATa/alpha scc3::NatMX4/WT, leu::Scc3 (K423E, K513E, K520E)-HA3::LEU* | This study | KN27696 | |
| Strain, strain background (*S. cerevisiae*) | *MATa scc3::NatMX4, Scc1-PK6::TRP1, leu::Scc3 (K423E, K513E, K520E)-HA3::LEU* | This study | KN27697 | |
| Strain, strain background (*S. cerevisiae*) | *MATa/alpha scc3::NatMX4/WT, leu::Scc3 (K224E, K225E, R226E, K423E, K513E, K520E)-HA3::LEU* | This study | KN27763 | |
| Strain, strain background (*S. cerevisiae*) | *MATa ura::ADH1promoter-OsTIR1-9myc::URA3, Scc3-PK3-aid::KanMX4, leu::Scc3-HA3::LEU* | This study | KN27796 | |
| Strain, strain background (*S. cerevisiae*) | *MATa ura::ADH1 promoter-OsTIR1-9myc::URA3, Scc3-PK3-aid::KanMX4 leu::Scc3 (K224E, K225E, R226E, K423E, K513E, K520E)-HA3::LEU* | This study | KN27802 | |
| Strain, strain background (*S. cerevisiae*) | *MATa ura::ADH1promoter-OsTIR1-9myc::URA3, Scc3-HA3-aid::KanMX4, Scc1-PK6::TRP1, leu::Scc3 (K224E, K225E, R226E, K423E, K513E, K520E)-HA3::LEU* | This study | KN27804 | |
| Strain, strain background (*S. cerevisiae*) | *MATa ura::ADH1 promoter-OsTIR1-9myc::URA3, Scc3-HA3-aid::KanMX4, Scc1-PK6::TRP1, leu::Scc3-HA3::LEU* | This study | KN27821 | |

*Continued on next page*

*Continued*

| Reagent type (species) or resource | Designation | Source or reference | Identifiers | Additional information |
|---|---|---|---|---|
| Strain, strain background (*S. cerevisiae*) | *MATa ura::ADH1 promoter-OsTIR1-9myc::URA3, Scc3-HA3-aid::KanMX4, leu::Scc3-HA3::LEU, Scc2-PK9::NatMX* | This study | KN28075 | |
| Strain, strain background (*S. cerevisiae*) | *MATa ura::ADH1 promoter-OsTIR1-9myc::URA3, Scc3-HA3-aid::KanMX4, Scc2-PK9::NatMX, leu::Scc3 (K224E, K225E, R226E, K423E, K513E, K520E)-HA3::LEU* | This study | KN28287 | |
| Strain, strain background (*S. frugiperda*) | Sf9 insect cells | ThermoFisher | Cat# 11496015 | |
| Antibody | Anti-His (mouse) | Sigma | Cat# SAB1305538-400UL | 1:2000 |
| Antibody | Anti-mouse HRP | ThermoFisher | Cat# 62–6520 | 1:5000 |
| Antibody | Anti-Smc3 (mouse) | Bethyl Laboratories | Cat# A300-060A | 1:500 |
| Antibody | Anti-Strep HRP | iba | Cat# 2-1502-001 | 1:4000 |
| Recombinant DNA reagent | pACEbac1 *SMC1-His* | This Study | | |
| Recombinant DNA reagent | pACEbac1 *SMC3* | This Study | | |
| Recombinant DNA reagent | pACEbac1 *SMC1-His SMC3* | This Study | | |
| Recombinant DNA reagent | pACEbac1 *Scc2$^{133-1493}$-2xStrepII* | This Study | | |
| Recombinant DNA reagent | pACEbac1 *SCC2-2xStrepII* | This Study | | |
| Recombinant DNA reagent | pACEbac1 *2xStrepII-Scc2$^{151-1493}$* | This Study | | |
| Recombinant DNA reagent | pACEbac1 *2xStrepII-SCC3* | This Study | | |
| Recombinant DNA reagent | pIDC *SCC1-2xStrepII* | This Study | | |
| Recombinant DNA reagent | pIDC *Scc1$^{269-451}$-2xStrepII* | This Study | | |
| Recombinant DNA reagent | pIDC *Scc1$^{150-298}$-2xStrepII* | This Study | | |
| Recombinant DNA reagent | pIDC *His-SCC4* | This Study | | |
| Chemical compound, drug | ATP Lithium Salt | Sigma | Cat# 11140965001 | |
| Chemical compound, drug | Bismaleimidoethane (BMOE) | ThermoFisher | Cat# 22323 | |
| Chemical compound, drug | Complete EDTA free protease inhibitor cocktail | Roche | Cat# 4693132001 | |

*Continued on next page*

*Continued*

| Reagent type (species) or resource | Designation | Source or reference | Identifiers | Additional information |
|---|---|---|---|---|
| Chemical compound, drug | Cre Recombinase | New England Biolabs | Cat# M0298S | |
| Chemical compound, drug | Desthiobiotin | Fisher Scientific | Cat# 12753064 | |
| Chemical compound, drug | EtBr | ThermoFisher | Cat# 15585011 | |
| Chemical compound, drug | Fetal Bovine Serum | Sigma | Cat# 12303C | |
| Chemical compound, drug | FuGENE HD Transfection reagent | Promega | Cat# E2311 | |
| Chemical compound, drug | Gibson Assembly Mix | New England Biolabs | Cat# E2611L | |
| Chemical compound, drug | Immobilon Western ECL | Millipore | Cat# WBLKS0500 | |
| Chemical compound, drug | NuPAGE 3–8% Tris-Acetate Protein Gels | ThermoFisher | Cat# EA0378BOX | |
| Chemical compound, drug | PMSF | Sigma | Cat# 329-98-6 | |
| Chemical compound, drug | Quick Coomassie Stain | Generon | Cat# GEN-QC-STAIN | |
| Chemical compound, drug | RNase A | Roche | Cat# 10109169001 | |
| Chemical compound, drug | Sf900 II SFM | ThermoFisher | Cat# 10902104 | |
| Chemical compound, drug | Supernuclease | SinoBiological | Cat# SSNP01 | |
| Chemical compound, drug | TCEP | ThermoFisher | Cat# 20490 | |
| Chemical compound, drug | 4xLDS | ThermoFisher | Cat# NP0007 | |
| Commercial assay or kit | HiLoad 16/60 Superdex 200 | GE Healthcare | Cat# GE28-9893-35 | |
| Commercial assay or kit | HiSpeed Plasmid Maxi Kit | Qiagen | Cat# 12663 | |
| Commercial assay or kit | HiTrap Q HP | GE Healthcare | Cat# GE29-0513-25 | |
| Commercial assay or kit | StrepTrap HP | Fisher Scientific | Cat# 11540654 | |
| Commercial assay or kit | Superose 6 Increase 10/300 GL | VWR | Cat# 29-0915-96 | |

*Continued on next page*

*Continued*

| Reagent type (species) or resource | Designation | Source or reference | Identifiers | Additional information |
|---|---|---|---|---|
| Commercial assay or kit | EnzChek phosphate assay kit | Invitrogen | Cat# E6646 | |
| Software, algorithm | RELION 3.1 | doi:10.1016/j.jsb.2012.09.006 | | |
| Software, algorithm | CtfFind4 | doi:10.1016/j.jsb.2015.08.008 | | |
| Software, algorithm | Warp | doi:10.1038/s41592-019-0580-y | | |
| Software, algorithm | CrYOLO 1.5 | doi:10.1038/s42003-019-0437-z | | |
| Software, algorithm | Chimera | https://www.cgl.ucsf.edu/chimera/ | | |
| Software, algorithm | ChimeraX 1.0 | https://www.cgl.ucsf.edu/chimerax/ | | |
| Software, algorithm | COOT | doi:10.1107/S0907444910007493 | | |
| Software, algorithm | MAIN | doi:10.1107/S0907444913008408 | | |
| Software, algorithm | Phenix.real_space_refinement | doi:10.1107/S2059798318006551 | | |
| Software, algorithm | PYMOL 2 | https://pymol.org/2/ | | |
| Software, algorithm | SWISS-MODEL | https://swissmodel.expasy.org | | |
| Other | Quantifoil Au 2/2 holely carbon 200 mesh cryoEM grids | Quantifoil GmbH | | |
| Other | Ultrafoil 2/2 holely gold 200 mesh cryoEM grids | Quantifoil GmbH | | |

## Recombinant yeast cohesin complex cloning

The *S. cerevisiae* genes *SMC1*, *SMC3*, *SCC3*, *SCC2*, *SCC1*, and *SCC4* were codon optimised for expression in *Spodoptera frugiperda* cells and synthesised using the Genescript Thermo Fisher service. These were then cloned into MultiBac vectors. Tag introduction and mutagenesis was achieved through Gibson assembly (New England Biolabs) to generate *SMC1*-His, 2xStrepII-*SCC3*, *SCC1*-2xStrepII, *SCC2*-2xStrepII, *SCC2*$^{133-1493}$-2xStrepII, and His-*SCC4*. *SMC3 SMC1*-His, 2xStrepII-*SCC3*, *SCC2*-2xStrepII, and *SCC2*$^{133-1493}$-2xStrepII were cloned into pACEbac1 vectors, and *SCC1*-2xStrepII and His-*SCC4* cloned into pIDC vectors. *SMC1*-His and *SMC3* were then combined into the same vector via cloning to create a pACEbac1 *SMC1*-His *SMC3*. Vectors containing cohesin trimers were generated by combining pACEbac1 *SMC1*-His *SMC3* with pIDC *SCC1*-2xStrepII by a Cre recombinase reaction (New England Biolabs). The vector for the Scc2/4 expression was also created by combining the pACEbac1 *SCC2*-2xStrepII with pIDC His-*SCC4* using Cre recombinase.

## Virus generation and protein expression

DNAs were first transformed into DH10Bac (Thermo Fisher) cells and bacmids containing the expression vector screened for by blue-white selection. DNA was then extracted and 2 µg of bacmid DNA was transfected into 2 ml *S. frugiperda* Sf9 cells (Thermo Fisher) at a cell density of $1 \times 10^6$ cells ml$^{-1}$ using FuGENE HD reagent (Promega), grown in Sf900 II SFM media (Thermo Fisher). These were then incubated at 27°C for 5 days to create P1 virus. P2 virus was then amplified by infecting 50 ml Sf9 cells at a density of $2 \times 10^6$ cells ml$^{-1}$ with 500 µl P1 virus and incubating in the dark at

27°C for 3 days with shaking at 100 rpm. P2 virus was then harvested by pelleting cells by centrifugation at 4000 g and decanting into 5% FBS (Sigma), and then stored in the dark at 4°C. Typically, proteins were then expressed by adding 5 ml P2 virus to 500 ml Sf900 cells at a density of $2 \times 10^6$ cells ml$^{-1}$ and incubating in the dark at 27°C for 2 days with shaking at 100 rpm. Cells were then harvested by centrifugation at 1000 g, washed with PBS, and then frozen in liquid nitrogen and stored at −80°C.

## Protein purification

Cells were thawed in Buffer A (50 mM HEPES pH 7.5, 150 mM NaCl, 1 mM TCEP (Thermo Fisher), 5% glycerol) supplemented with 1 Complete Protease Inhibitor (EDTA-free) tablet (Roche), 70 µg RNAse A (Roche), and 100 U ml$^{-1}$ Supernuclease (Sino Biological) and then lysed by sonication. Following sonication, cell lysate was supplemented with 1 mM PMSF (Sigma). Proteins were then purified via a three strep purification protocol. First, proteins were purified via affinity pulldown of their StrepII tags using a StrepTrap HP column (Fisher Scientific) and eluted into Buffer A supplemented with 2.5 mM desthiobiotin (Fisher Scientific). Scc2 constructs were eluted into 50 mM Tris pH 8.0 rather than 50 mM HEPES pH 7.5. Proteins were then further purified by anion exchange chromatography using a 5 ml HiTrap Q HP column (GE Healthcare) across a gradient of 100 mM to 1M NaCl. Scc2 constructs were eluted across a gradient of 0 mM to 1 M NaCl. Finally, proteins were purified via size exclusion chromatography using a Superose 6 increase 10/300 GL column (VWR) for cohesin trimers and a HiLoad 16/600 Superdex 200 column (GE Healthcare) for Scc3 and the Scc2 constructs.

## Purification of pUC19 plasmid DNA

pUC19 plasmid was transformed into TOP10 (Thermo Fisher) cells and grown overnight at 37°C. The next day a single colony was inoculated into 250 ml SOB++ media and grown at 37°C overnight for 16 hr. DNA was then purified via MaxiPrep (Qiagen) using precooled reagents and equipment and eluted into 50 mM HEPES pH 7.5. DNA was then further purified by CsCl$_2$ density gradient centrifugation in the presence of EtBr (Thermo Fisher). The DNA was then extracted and the EtBr removed by washing several times with butanol saturated with 50 mM HEPES pH 7.5 and then the butanol phase discarded. The CsCl$_2$ was then removed by dialysis against 2 L 50 mM HEPES pH 7.5 buffer over 24 hr at 4°C, with two buffer changes. The DNA was then collected and stored at −20°C.

## Protein gel electrophoresis and western blotting

Samples were mixed with 4xLDS sample buffer (Thermo Fisher), loaded onto a 3–8% Tris-acetate gel (Thermo Fisher) and separated at 150 V for 50 min. Gels were then either stained with Quick Coomassie stain (Generon) or transferred onto a 0.2 µm nitrocellulose membrane using a Trans-blot Turbo transfer pack (Bio-Rad). The antibodies used for western blotting were anti-His (Sigma), anti-Strep HRP conjugated (iba) and anti-Smc3 (Bethyl Laboratories). Primary antibodies were probed with anti-mouse HRP conjugated antibodies (Thermo Fisher).

## Protein crosslinking assay

For protein crosslinking assays, 10 µl reactions were prepared containing 570 nM protein (buffered with 50 mM HEPES pH 7.5, 50 mM NaCl, 5 mM MgCl$_2$, 1 mM TCEP, 5% glycerol). If added, ATP (Sigma) and pUC19 were added to a concentration of 5 mM and 60 nM respectively. Reactions mixes were first incubated on ice for 5 min before adding 1 µl BMOE (Thermo Fisher) to a final concentration of 0.64 mM. Reactions were then incubated on ice for 6 min. Samples were denatured by adding 4xLDS buffer and heating at 70°C for 10 min before being separated in 3–8% Tris-acetate gels (Thermo Fisher) run at 150 V for 3 hr. Gels were stained with Quick Coomassie stain (Generon).

## DNA entrapment assay

For DNA entrapment assays, 13 µl reactions were prepared containing 165 nM protein and 9.3 nM supercoiled pUC19 (buffered with 50 mM HEPES pH 7.5, 20 mM NaCl, 1 mM MgCl$_2$, 5% glycerol). Scc2C was added to a concentration of 55 nM. These were incubated on ice for 5 min before reactions were initiated by addition ATP (Sigma) to a final concentration of 5 mM (1 µl ATP added to 12 µl protein DNA mix). Typically, reactions were then incubated at 24°C for either 40 min or 2 min

depending on the compartment being assessed. To these, 1.5 μl BMOE (Thermo Fisher) was added to a final concentration of 0.64 mM and samples were incubated on ice for 6 min. Samples were then denatured by addition of 1.5 μl 10% SDS and heated at 70°C for 20 min. DNA loading dye was added and mixtures separated in a 0.8% agarose gel, run at 50 V for 17 hr at 4°C. The gel was stained with EtBr (Thermo Fisher) and visualised by UV light. Images shown are representative of 2 independent experiments.

## ATPase assay
ATPase experiments were carried out as described in *Petela et al., 2018*.

## Calibrated ChIP-sequencing
ChIP sequencing experiments were carried out as described in *Petela et al., 2018*.

## Electromobility shift assay (EMSA)
A FAM-labelled 39 base pair HPLC purified DNA oligo (Invitrogen, GAATTCGGTGCGCATAATGTA TATATTATGTTAAATAAGCTT) was annealed to a complementary DNA oligo to form dsDNA by heating to 95°C for 5 min and then decreasing the temperature to 4°C in 0.1°C increments (buffered in 100 mM potassium acetate, 50 mM HEPES pH 7.5) to a final concentration of 45 μM. The reactions were then prepared by adding 0.3 μM FAM-dsDNA to increasing concentrations of protein (buffered in 50 mM HEPES pH 7.5, 75 mM NaCl, 1 mM TCEP, 10% glycerol) in a final volume of 10 μL. The reaction mix was then incubated on ice for 30 min in the dark. These were then separated in 5% acrylamide gels prepared with 0.5% TAE (40 mM Tris, 20 mM acetic acid, 1 mM EDTA) run at 100 V for 1 hr at 4°C in the dark. FAM-labelled DNA was visualised directly on a Fujifilm FLA7000 scanner with the LD473/Y[520] filter.

## Cell viability of Scc3 mutants
Mutant *scc3* alleles (under their native promoter) were incorporated at the *leu2* locus in heterozygous *SCC3/Δscc3* cells. Diploids were sporulated and tetrads dissected onto YPD plates. The genotype of the resulting haploids was determined by replica plating. All mutations were confirmed by DNA sequencing.

## Cryo-EM sample preparation
1 μM purified *S. cerevisiae* cohesin EQ/EQ trimer (Smc1E1158Q, Smc3E1155Q, Scc1-2xStrepII) was incubated for 30 min at 4 °C with 1 μM Scc2C2(151–1493), forming cohesin tetramer, 5 mM ATP, and 1.3 μM of a 40 bp dsDNA (5'-GAATTCGGTGCGCATAATGTATATTATGTTAAATAAGCTT-3', 5'-AAGCTTATTTAACATAATATACATTATGCGCACCGAATTC-3') or relaxed plasmid DNA. The plasmid was 1789 bp, derived from pUC19, containing a single site for Nt.BspQI nicking endonuclease. Nicking was performed according to the manufacturer's instructions for 60 min at 50°C (NEB) and the product was purified using a PCR purification kit (Qiagen) and eluted in water. For vitrification, 3 μl of sample were applied to glow-discharged Quantifoil Au 2/2 holey carbon 200 mesh grids or Ultrafoil Au 2/2 holey gold 200 mesh grids (Quantifoil), and flash frozen in liquid ethane using an FEI Vitrobot Mark IV (Thermo Fisher Scientific) and a liquid-ethane cryostat set to −180°C (*Russo et al., 2016*).

## Cryo-EM data collection
All cryo-EM images were collected on a Titan Krios electron microscope operated at 300 kV (Thermo Fisher Scientific). Images for the cohesin tetramer:40 bp dsDNA complex were collected with a Quantum energy filter (GIF) in front of a K3 Summit direct electron camera in super-resolution mode (both Gatan). The nominal defocus range was set to 1.5 ~ 3.3 μm. Each image was dose-fractionated over 55 frames with a dose rate of 1 electrons per Å per image. A total 11,944 micrographs were collected in three separate sessions using Krios III at LMB (calibrated pixel size: 1.069 Å / pixel at nominal magnification of 81,000) and the Krios microscope at the Biochemistry Department at the University of Cambridge (pixel size: 1.07 Å / pixel at nominal magnification of 81,000). Because the images showed strong orientation bias, 6580 micrographs of the datasets were collected at tilts of 25° or 30°. For the tetramer:plasmid DNA complex, 535 images were collected using the Volta phase

plate (VPP) (*Danev and Baumeister, 2016*) and a Gatan K2 Summit direct electron camera in counting mode on Krios II at LMB (calibrated pixel size: 1.00 Å / pixel at magnification of 105,000) with total doses of 45 electrons per Å², dose fractionated into 40 movie frames using a nominal defocus range of 0.6 ~ 1.0 μm.

## Cryo-EM data processing and reconstruction

RELION 3.1 was used for all data processing unless otherwise specified (*Scheres, 2012*). The resolution was determined based on gold standard Fourier shell correlation (FSC) using the 0.143 criterion (*Rosenthal and Henderson, 2003*). Using RELION's own motion correction implementation, movie frames were aligned and combined with dose weighting using $7 \times 5$ patches or $5 \times 5$ patches for K3 and K2 datasets, respectively. CTF parameters were estimated with CtfFind4 (*Rohou and Grigorieff, 2015*) for un-tilted images. Focus-gradient patch CTF estimation was performed for tilted images using the programme Warp (*Tegunov and Cramer, 2019*). For particle picking, RELION and crYOLO were used (*Wagner et al., 2019*).

For the cohesin tetramer:40 bp DNA complex, initially particles were picked from a subset of ~1000 images with a Laplacian-of-Gaussian blob as a template using RELION, followed by particle extraction and reference-free 2D classification. An initial 3D model was obtained using particles from selected 2D class images showing different orientations. Then, particle coordinates from images that formed good 2D classes were used to train a model in crYOLO, and particle picking was performed with the trained model. Picked particles were extracted using a box size of $320^2$ pixels followed by 3D classification. After several rounds of 3D classification, 588,164 particles with clear density for the Smc1/3 heads and Scc2 were selected and 3D refined to 3.8 Å resolution. Further CTF refinement and Bayesian polishing were performed in RELION followed by another round of 3D auto-refinement, which resulted in a final 3.4 Å map of the cohesin Smc1EQ, Smc3EQ, Scc1, Scc2, ATP, 40 bp DNA complex showing the head part only.

For processing of the tetramer:plasmid DNA complex, picked particles from crYOLO were binned and extracted in a box of $160^2$ pixels (2 Å per pixel), followed by several rounds of 2D/3D classifications with the map of the tetramer:40 bp DNA complex as initial 3D model. 23,704 particles showing the head part clearly were selected and re-extracted using a box size of $320^2$ pixels (1.07 Å/pix) and were 3D refined to 7.3 Å resolution.

In order to obtain a map of the entire complex, particles from the first 3D classification during the processing of the tetramer:40 bp DNA complex (above) containing clear DNA density were re-centred on the joint region of the complex and re-extracted in a box size of $320^2$ pixels (2 Å per pixel). After initial 2D classification, an initial model was generated with a few class averages in different orientations, followed by 3D classification. After several rounds of 3D classification, 21,343 particles containing well-ordered coiled coil and hinge density were 3D refined to 10 Å resolution.

## Model building

A homology model of yeast Scc2 was obtained from SWISS-MODEL (*Waterhouse et al., 2018*) using a crystal structure of Scc2 from *E. gossypii* (PDB 5ME3) as the template (*Chao et al., 2017*). Crystal structures of yeast Smc1 head (PDB 1W1W; *Haering et al., 2004*) and Smc3 head (PDB:4U × 3; *Gligoris et al., 2014*) and the Scc2 homology model were docked into the tetramer:40 bp DNA cryo-EM density map using UCSF Chimera X (*Goddard et al., 2018*). MAIN (*Turk, 2013*) and COOT (*Emsley et al., 2010*) were used for manual rebuilding, followed by refinement using Phenix.real_space_refinement (*Afonine et al., 2018*). Manual-rebuilding and refinement were repeated for several cycles. The cryo-EM map and the atomic model of the cohesin head segment at 3.4 Å resolution (*Supplementary file 1*) were deposited in the EM Data Bank (EMDB) and Protein Data Bank (PDB) with accession numbers EMD-11585 and PDB 6ZZ6.

## Acknowledgements

We thank all members of the Nasmyth and Löwe groups for valuable discussions and many unseen contributions over the course of this work. We would like to thank Nicolas Jean (MRC-LMB, Cambridge, UK) for a sample of the relaxed plasmid DNA, Yang Lee (MRC-LMB) for help with tilted data processing, Dima Chirgadze and Steven W Hardwick (Biochemistry Department, Cambridge

University, UK) and all the staff of the MRC-LMB EM Facility for help with cryo-EM data collection, and Jake Grimmett and Toby Darling for supporting scientific computing (MRC-LMB).

## Additional information

### Funding

| Funder | Grant reference number | Author |
|---|---|---|
| Wellcome | 107935/Z/15/Z | Kim A Nasmyth |
| Cancer Research UK | 26747 | Kim A Nasmyth |
| Medical Research Council | U105184326 | Jan Löwe |
| Wellcome | 202754/Z/16/Z | Jan Löwe |

The funders had no role in study design, data collection and interpretation, or the decision to submit the work for publication.

### Author contributions

James E Collier, Conceptualization, Data curation, Formal analysis, Validation, Investigation, Visualization, Methodology, Writing - original draft, Writing - review and editing; Byung-Gil Lee, Data curation, Software, Formal analysis, Visualization, Methodology, Writing - review and editing; Maurici Brunet Roig, Stanislav Yatskevich, Investigation, Methodology, Writing - review and editing; Naomi J Petela, Formal analysis, Investigation; Jean Metson, Menelaos Voulgaris, Investigation; Andres Gonzalez Llamazares, Visualization, Writing - review and editing; Jan Löwe, Resources, Supervision, Funding acquisition, Visualization, Methodology, Writing - review and editing; Kim A Nasmyth, Conceptualization, Supervision, Funding acquisition, Writing - original draft, Writing - review and editing

### Author ORCIDs

James E Collier https://orcid.org/0000-0002-9904-9423
Byung-Gil Lee https://orcid.org/0000-0001-9565-6114
Andres Gonzalez Llamazares https://orcid.org/0000-0001-5404-6360
Jan Löwe https://orcid.org/0000-0002-5218-6615
Kim A Nasmyth https://orcid.org/0000-0001-7030-4403

### Decision letter and Author response

Decision letter https://doi.org/10.7554/eLife.59560.sa1
Author response https://doi.org/10.7554/eLife.59560.sa2

## Additional files

### Supplementary files

- Supplementary file 1.
- Transparent reporting form

### Data availability

ChIP-seq data has been deposited to GEO with accession number GSE156616. The cryo-EM data have been deposited in PDB under the accession code 6ZZ6 and in EM Data Bank under the accession code EMD-11585.

The following datasets were generated:

| Author(s) | Year | Dataset title | Dataset URL | Database and Identifier |
|---|---|---|---|---|
| Petela NJ, Nasmyth KA | 2020 | Transport of DNA within cohesin involves clamping on top of engaged heads by Scc2 and | https://www.ncbi.nlm.nih.gov/geo/query/acc.cgi?acc=GSE156616 | NCBI Gene Expression Omnibus, GSE156616 |

entrapment within the ring by Scc3

| Lee B-G, GonzalezLlamazares A, Collier J, Nasmyth KA, Löwe J | 2020 | Cryo-EM structure of *S.cerevisiae* cohesin-Scc2-DNA complex | https://www.rcsb.org/structure/6ZZ6 | RCSB Protein Data Bank, 6ZZ6 |
| Lee B-G, GonzalezLlamazares A, Collier J, Nasmyth KA, Löwe J | 2020 | Cryo-EM structure of *S.cerevisiae* cohesin-Scc2-DNA complex | https://www.ebi.ac.uk/pdbe/entry/emdb/EMD-11585 | Electron Microscopy Data Bank, EMD-11585 |

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
