## [Decision Letter]

**Acceptance summary:**

This paper provides compelling evidence for DNA entrapment by cohesin. Using cleverly-designed cross-linking experiments and cryo-EM, the authors show that Scc2 clamps DNA into a cohesin sub-compartment, which likely precedes entrapment within a larger compartment, in a reaction dependent on Scc3. These two modes of entrapment may represent cohesin states active in loop extrusion and sister chromatid cohesion, respectively.

**Decision letter after peer review:**

Thank you for submitting your article "Scc2 and Scc3 promote distinct modes of topological association between cohesin and DNA" for consideration by *eLife*. Your article has been reviewed by three peer reviewers, including Adèle L Marston as the Reviewing Editor and Reviewer #1, and the evaluation has been overseen by Jessica Tyler as the Senior Editor. The following individuals involved in review of your submission have agreed to reveal their identity: Hongtao Yu (Reviewer #2); Daniel Panne (Reviewer #3).

The reviewers have discussed the reviews with one another and the Reviewing Editor has drafted this decision to help you prepare a revised submission.

Summary:

Cohesin can associate with DNA in a topological and non-topological manner which may underlie its roles in sister chromatid cohesion and DNA loop extrusion. How cohesin loads onto and entraps DNA is an important question. Until now, topological binding has not been demonstrated in vitro. This manuscript uses circular minichromosomes, purified proteins and a cleverly designed in vitro crosslinking assay to examine the topology of DNA-cohesin interactions. Cohesin can form several different compartments within which DNA could potentially be trapped. Using pairs of cysteine crosslinks at these interfaces Collier et al. measure the impact of closure of these interfaces on topological DNA binding. They show for the first time in vitro that topological binding does indeed occur. They convincingly demonstrate that DNA binding by Scc3 and Scc2 as well as ATP stimulate such a DNA entry reaction. They show that Scc3 and its DNA-binding ability are indispensable for the entrapment of DNA within cohesin's S-K chamber, with Scc2 merely acting as a stimulator. In contrast, Scc2 and its DNA-binding ability are crucial for fast DNA entrapment within E-S and E-K compartments, whereas Scc3 is dispensable. The evidence presented is compelling and provides experimental support for the DNA entrapment state captured by recent cryoEM structures of DNA-bound human and fission yeast cohesin and NIPBL/Scc2 complexes. The discovery of different roles of Scc2 and Scc3 in DNA entrapment by cohesin is significant. This is an impressive study which provides a major step forward in addressing the long standing question in the field as to how DNA entrapment is achieved by the cohesin complex. It also raises many further interesting questions for future study. A few points should be addressed.

Essential revisions:

1) If entrapment is indeed topological and not a result of simple physical association, one would expect that no binding by 6C cohesin is seen on linearized DNA? As others have used salt sensitivity as a measure of topological loading, it would be important to confirm this point. This would be an additional confirmation of the system, but, as a 5C control shows no DNA binding, is not a 'sina que non'.

2) Figure 2: It is not clear why 2 repeat experiments +Scc2/+Scc3 are shown while a negative control -Scc2/-Scc3 is missing. It would be good to know if the residual activity seen in +Scc2/-Scc3 disappears in the absence of Scc2.

3) Figure 2: The Smc1E1158Q Smc3E1155Q mutant may have residual ATP hydrolysis activity. What happens when a non-hydrolysable analogue (with the E->Q mutant) is used or Mg^2+^ cofactor is omitted etc? These should support entrapment if ATP hydrolysis is not required for entrapment?

4) Figure 6: If entrapment of DNA within the E-S compartments involves dissociation of the coiled coils, one would predict that crosslinking between Smc1K201C and Smc3K198C is precluded in the presence of DNA. To investigate this, can they test the effect of ATP, Scc2 and DNA on coiled coil crosslinking? Direct demonstration would support the notion that head engagement disrupts coiled coil interactions.

5) Figure 6B: If ATP binding and adoption of the E state disrupts coiled coil alignment, is it not surprising to see that Smc1K201C and Smc3K198C efficiently cross link in the presence of ATP? Is it not likely that these coiled coil interactions remain somewhat dynamic, even in the presence of ATP, thus allowing Smc1K201C and Smc3K198C crosslinking?

6) Figure 6D. ATP (when added together with Scc2 and DNA) causes mild reduction in J-state crosslinking in Figure 6D. Did authors test the effect of a non-hydrolyzable ATP analogue?

7) Figure 7A: If the presented model is correct, one possibility is that non-topological binding is converted transiently into topological binding by SMC head engagement. That is, cohesin could switch between non-topological and topological binding modes as a function of head engagement?

---

## [Author Response]

Essential revisions:1) If entrapment is indeed topological and not a result of simple physical association, one would expect that no binding by 6C cohesin is seen on linearized DNA? As others have used salt sensitivity as a measure of topological loading, it would be important to confirm this point. This would be an additional confirmation of the system, but, as a 5C control shows no DNA binding, is not a 'sina que non'.

We have provided further evidence that the interaction we are describing is indeed topological in nature by including an experiment showing that entrapment due to BMOE-mediated DNA-protein catenations is specific to circular DNA, with linear DNA unable to form these catenations (Figure 1—figure supplement 1D).

2) Figure 2: It is not clear why 2 repeat experiments +Scc2/+Scc3 are shown while a negative control -Scc2/-Scc3 is missing. It would be good to know if the residual activity seen in +Scc2/-Scc3 disappears in the absence of Scc2.

We have discussed in the text that levels of entrapment in the absence of Scc2 and Scc3 is comparable to those seen in the presence of Scc2 but the absence Scc3.

3) Figure 2: The Smc1E1158Q Smc3E1155Q mutant may have residual ATP hydrolysis activity. What happens when a non-hydrolysable analogue (with the E->Q mutant) is used or Mg^2+^ cofactor is omitted etc? These should support entrapment if ATP hydrolysis is not required for entrapment?

In respect to using ATP analogues, we found that their use resulted in considerable reductions in head engagement, as measured by our site-specific cysteine crosslinking. This is not to say that head engagement did not take place per se and could simply mean that minor structural changes abrogated head engagement as measured by our crosslink pair. In light of this, we decided not to pursue the related experiments further.

4) Figure 6: If entrapment of DNA within the E-S compartments involves dissociation of the coiled coils, one would predict that crosslinking between Smc1K201C and Smc3K198C is precluded in the presence of DNA. To investigate this, can they test the effect of ATP, Scc2 and DNA on coiled coil crosslinking? Direct demonstration would support the notion that head engagement disrupts coiled coil interactions.

We carried out the suggested crosslinking experiments studying the interactions between the coiled coil under conditions where head engagement takes place, and conditions that lead to entrapment of DNA in the E-S compartment. We failed to find a condition where we saw significant reduction in crosslinking around this region and have discussed this in the text. Perhaps any occupancy of DNA within the E-S compartment is transient making disruption of the coiled coils difficult to measure by crosslinking. It is possible to see disruption however when we impose the constraint that DNA must be present and that is the power of the experiment that we describe.

5) Figure 6B: If ATP binding and adoption of the E state disrupts coiled coil alignment, is it not surprising to see that Smc1K201C and Smc3K198C efficiently cross link in the presence of ATP? Is it not likely that these coiled coil interactions remain somewhat dynamic, even in the presence of ATP, thus allowing Smc1K201C and Smc3K198C crosslinking?

As we are able to see simultaneous crosslinking between this coiled coil cysteine pair and the engaged heads, we agree with the suggestion that this region of the coiled coil must be highly dynamic.

6) Figure 6D. ATP (when added together with Scc2 and DNA) causes mild reduction in J-state crosslinking in Figure 6D. Did authors test the effect of a non-hydrolyzable ATP analogue?

See earlier point regarding ATP analogues.

7) Figure 7A: If the presented model is correct, one possibility is that non-topological binding is converted transiently into topological binding by SMC head engagement. That is, cohesin could switch between non-topological and topological binding modes as a function of head engagement?

This is not possible. In order for DNAs to enter the S-K ring, it must pass through one of its three interfaces. Merely changing the engagement of the heads cannot achieve this.